# Voltage-clamp fluorometry analysis of structural rearrangements of ATP-gated channel P2X2 upon hyperpolarization

**Rizki Tsari Andriani[1,2]\*, Yoshihiro Kubo[1,2]\***

[1]Division of Biophysics and Neurobiology, National Institute for Physiological Sciences, Aichi, Japan; [2]Department of Physiological Sciences, The Graduate University for Advanced Studies, School of Life Science, Kanagawa, Japan

**Abstract** Gating of the ATP-activated channel P2X2 has been shown to be dependent not only on [ATP] but also on membrane voltage, despite the absence of a canonical voltage-sensor domain. We aimed to investigate the structural rearrangements of rat P2X2 during ATP- and voltage-dependent gating, using a voltage-clamp fluorometry technique. We observed fast and linearly voltage-dependent fluorescence intensity (F) changes at Ala337 and Ile341 in the TM2 domain, which could be due to the electrochromic effect, reflecting the presence of a converged electric field. We also observed slow and voltage-dependent F changes at Ala337, which reflect structural rearrangements. Furthermore, we determined that the interaction between Ala337 in TM2 and Phe44 in TM1, which are in close proximity in the ATP-bound open state, is critical for activation. Taking these results together, we propose that the voltage dependence of the interaction within the converged electric field underlies the voltage-dependent gating.

**\*For correspondence:**
kiki@nips.ac.jp (RTA);
ykubo@nips.ac.jp (YK)

**Competing interests:** The authors declare that no competing interests exist.

## Introduction

P2X2 is a member of the P2X receptor family, a ligand-gated cation channel that opens upon the binding of extracellular ATP (*Brake et al., 1994*; *Valera et al., 1994*). P2X receptors consist of seven sub-classes (P2X1–P2X7) that assemble to form trimeric homomers or heteromers (e.g., P2X2/P2X3) (*Radford et al., 1997*; *North, 2002*; *Jiang et al., 2003a*). Based on known crystal structures, P2X receptors have two transmembrane (TM) domains (TM1 and TM2), a large extracellular ligand binding loop (ECD), which is the location of the ATP binding site, and intracellular N- and C- termini (*Kawate et al., 2009*; *Hattori and Gouaux, 2012*; *Mansoor et al., 2016*; *McCarthy et al., 2019*).

P2X2 is mainly distributed in smooth muscle, the central nervous system (CNS), the retina, chromaffin cells, and autonomic and sensory ganglia (*Burnstock, 2003*). Recent studies have shown that the P2X2 receptor in hair cells and supporting cells has an important role in auditory transduction. A dominant negative polymorphism in human results in progressive hearing loss (*Yan et al., 2013*). Furthermore, P2X2 in the cochlea is involved in adaptation to elevated sound levels (*Housley et al., 2013*).

The P2X2 receptor has complex gating properties that consist of (1) [ATP]-dependent gating and (2) voltage-dependent gating. The latter is despite the absence of a canonical voltage sensor domain, in clear contrast to typical voltage-gated ion channels. In the presence of ATP, there is a gradual increase of inward current upon hyperpolarization. ATP shifts the conductance–voltage relationship toward depolarized potentials. Thus, activation of the P2X2 channel is voltage-dependent as well as [ATP]-dependent (*Nakazawa et al., 1997*; *Zhou and Hume, 1998*; *Nakazawa and Ohno, 2005*; *Fujiwara et al., 2009*; *Keceli and Kubo, 2009*). Previous studies have reported that this activation is indeed an intrinsic property of the channel (*Nakazawa et al., 1997*; *Zhou and Hume, 1998*; *Fujiwara et al., 2009*).

It is of interest to know how P2X2 has voltage-dependent gating despite the absence of a canonical VSD. Previous studies have extensively investigated the roles of amino acid residues in TM1 and TM2 during ATP-dependent gating and permeation (*Haines et al., 2001*; *Jiang et al., 2001*; *Li et al., 2004*; *Khakh and Egan, 2005*; *Cao et al., 2007*; *Samways et al., 2008*; *Cao et al., 2009*). In contrast, there is limited information about the roles of amino acid residues, particularly those within TM domains, in voltage-dependent gating. A previous study identified positively charged amino acid residues in the ATP binding pocket (K69, K71, R290, and K308; rP2X2 numbering) and aromatic amino acid residues in TM1 (Y43, F44, and Y47; rP2X2 numbering), which are critical for ATP- and voltage-dependent gating of the P2X2 receptor (*Keceli and Kubo, 2009*). However, those residues were not the only determinants of [ATP]- and voltage-dependent gating of the P2X2 receptor. The interpretation as to the mechanism is not yet straightforward and, thus, the key amino acid residue in the voltage sensing mechanism of the P2X2 receptor is yet to be discovered.

Moreover, the details of the structural rearrangements upon ATP binding in the pore region remain controversial. First, there are discrepancies between zfP2X4 structural data (*Kawate et al., 2009*; *Hattori and Gouaux, 2012*) and experimental data on P2X from metal bridging experiments, molecular dynamics simulations, and photo-switchable cross-linker experiments focusing on the TM domains (*Kracun et al., 2010*; *Li et al., 2010*; *Heymann et al., 2013*; *Habermacher et al., 2016*). Second, there are discrepancies between the crystal structures of the TM domains of zfP2X4 and hP2X3; hP2X3 has longer TM domains. Furthermore, the ATP-bound open state of hP2X3 and rP2X7 has a cytoplasmic cap that was not seen in the crystal structure of zfP2X4 (*Kawate et al., 2009*; *Hattori and Gouaux, 2012*; *Mansoor et al., 2016*; *McCarthy et al., 2019*). Thus, the present study aims at analyzing the structural rearrangements of the P2X2 receptor in (1) ATP- and (2) voltage-dependent gating, by voltage-clamp fluorometry (VCF), using a fluorescent unnatural amino acid (fUAA) as a probe.

The combination of fluorometry and voltage-clamp recording offers a powerful method for analyzing real-time conformational changes within the ion channel structure (*Mannuzzu et al., 1996*; *Cha and Bezanilla, 1997*; *Pless and Lynch, 2008*; *Nakajo and Kubo, 2014*; *Talwar and Lynch, 2015*). The use of an fUAA as a probe makes it possible to label any residue within the protein, including those in the lower TM and intracellular regions, which are not accessible by conventional VCF fluorophores, such as Alexa-488 maleimide (*Kalstrup and Blunck, 2013*; *Sakata et al., 2016*; *Kalstrup and Blunck, 2018*; *Klippenstein et al., 2018*). Moreover, direct incorporation of the fUAA increases the labeling efficiency and also prevents non-specific labeling (*Kalstrup and Blunck, 2013*; *Sakata et al., 2016*).

The fUAA used here, *3-(6-acetylnaphthalen-2-ylamino)−2-aminopropionic acid* (Anap), was incorporated into the rP2X2 protein by using a nonsense suppression method, in which the tRNA Anap-CUA and tRNA-synthetase pair is used to introduce Anap at an amber nonsense codon mutation (*Lee et al., 2009*; *Chatterjee et al., 2013*; *Klippenstein et al., 2018*), as shown in *Figure 1A*. By performing VCF recording, using Anap as a fluorophore, we analyzed the structural dynamics of the P2X2 receptor undergoing complex gating. In the present study, we observed evidence of voltage-dependent conformational changes in the transmembrane regions. We also investigated the key amino acid residues in each TM region whose interaction might have major contributions to the ATP- and voltage-dependent gating of the P2X2 receptor.

## Results

### Fluorescence signal changes of Anap-labeled P2X2 receptor evoked by ATP and voltage

As the P2X2 receptor does not have a canonical voltage-sensing domain (VSD), we performed Anap scanning by introducing TAG mutations one at a time in all regions of the P2X2 receptor, including the cytoplasmic N-terminus (eight positions), TM1 (20 positions), ECD, where the ATP binding site is located (25 positions), TM2 (24 positions), and cytoplasmic C-terminus (19 positions) (*Figure 1B,C*). The whole of TM1 and TM2 was scanned, as these are the transmembrane domains in which a non-canonical voltage sensor might be located.

Of the total of 96 positions of Anap mutants in the P2X2 receptor, many showed ATP-evoked fluorescence intensity (F) changes (ΔF) (*Supplementary file 1*). As major and overall structural

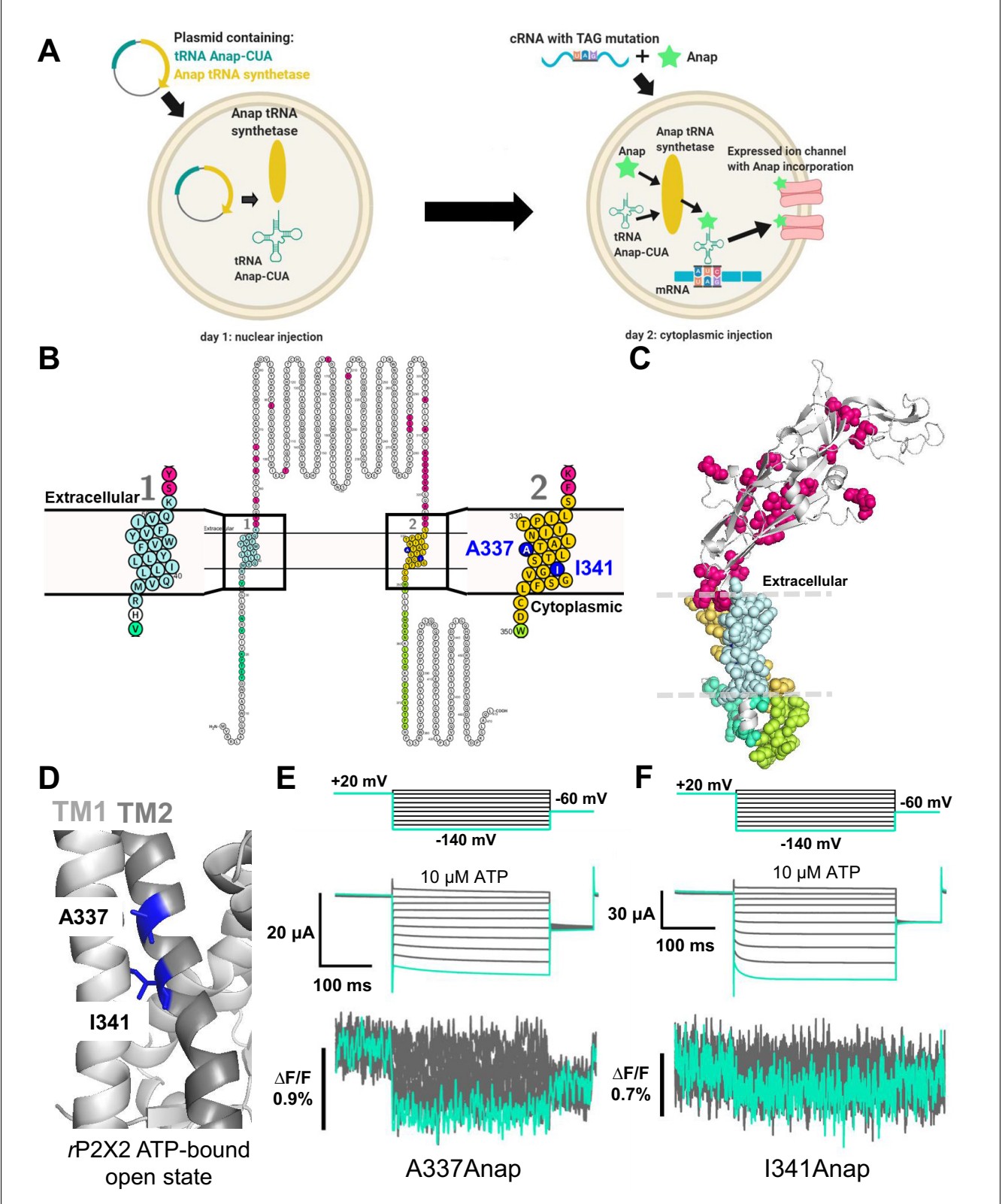

**Figure 1.** Fluorescence signal changes of Anap-incorporated P2X2 receptor evoked by ATP and voltage. (**A**) A scheme depicting the principle of the direct incorporation of fUAA (Anap) into the ion channel protein. The plasmid containing tRNA Anap-CUA and tRNA synthase is injected into the nucleus of *Xenopus laevis* oocytes. On the following day, channel cRNA with a TAG mutation is co-injected with Anap into the cytoplasm of the oocytes. Anap-incorporated channel protein was expressed successfully after the optimum incubation period. (**B, C**) A scheme to visualize the Anap

*Figure 1 continued on next page*

*Figure 1 continued*

scanning regions by individual amino acid residue representation (B) and within the protein structure (C), respectively. Anap mutant scanning was done by introducing TAG mutation one at a time in all regions of the P2X2 receptor (a total of 96 positions), which include the N-terminus (eight positions, turquoise), TM2 (24 positions, yellow), extracellular domain (ECD, 25 positions, magenta), TM1 (20 positions, light blue), and C-terminus (19 positions, lime green). Voltage-dependent fluorescence changes of Anap were observed only at A337 and I341 in the TM2 domain (colored by dark blue). (D) The sites of the introduced TAG mutations, A337 and I341 in the TM2 domain, which gave voltage-evoked fluorescence changes. All of *r*P2X2 structure representations in (C) and (D) were based on homology modeling from the ATP-bound open state *h*P2X3 crystal structure data (PDB ID: 5SVK; *Mansoor et al., 2016*). (E, F) Representative current traces and fluorescence signal upon ATP and voltage application in Anap mutants (A337: ΔF/F = 0.5 ± 0.2% at 440 nm [n = 3]; I341: ΔF/F = 0.3 ± 0.2% at 440 nm [n = 3], respectively). Source data are provided in *Figure 1—source data 1*.

The online version of this article includes the following source data and figure supplement(s) for figure 1:

**Source data 1.** Fluorescence signal changes of A337Anap and I341Anap evoked by ATP and voltage.
**Figure supplement 1.** ATP-evoked currents of A337Anap and I341Anap were inhibited by P2X2 non-specific blockers: Suramin and PPADS.
**Figure supplement 1—source data 1.** ATP-evoked currents of A337Anap and I341Anap were inhibited by Suramin and PPADS.
**Figure supplement 2.** Anap mutant scanning in the TM1 domain.
**Figure supplement 3.** Anap mutant scanning in the TM2 domain.

movement occurs upon the binding of ATP during the channel's transition from closed to open state in the P2X receptor (*Kawate et al., 2009*; *Hattori and Gouaux, 2012*; *Mansoor et al., 2016*; *McCarthy et al., 2019*), the results accord well with the expectation that an ATP-evoked fluorescence change would be observed at many positions labeled by Anap.

In contrast, out of 96 mutants tested, only two produced fluorescence intensity changes in response to voltage steps (*Figure 1E,F*; *Supplementary file 1*). These two positions are in TM2: A337 (ΔF/F = 0.5 ± 0.2% upon voltage change from +40 mV to −140 mV at 440 nm [n = 3], *Figure 1D,E*) and I341 (ΔF/F = 0.3 ± 0.2% upon voltage change from +40 mV to −140 mV at 440 nm [n = 3], *Figure 1D,F*). ATP-evoked currents of both constructs were inhibited by the P2X2 receptor non-specific blockers, Suramin and PPADS (*Figure 1—figure supplement 1*), confirming that the currents are indeed P2X2 receptor currents. Voltage-dependent F changes could not be detected at other scanned positions in TM1, TM2, or other regions (*Figure 1—figure supplements 2* and *3*; *Supplementary file 1*).

Although Anap ΔF was observed in several mutants, there are two major concerns, as follows: (1) ΔF is close to the limit of detection because signal-to-noise ratio is low, making it hard to perform further analysis, for example, ΔF-F-V relationships. (2) The incidence of fluorescence change detection in each batch is also low, 14.3 ± 4.1% (n = 5–16) and 16.02 ± 0.6% (n = 6–13) for A337Anap and I341Anap, respectively. Three out of 13 batches showed F change for A337Anap and 2 out of 10 batches showed F change for I341Anap. Thus, at this point, further analysis to determine the structural rearrangements with which Anap ΔF is associated could not be performed.

## SIK inhibitor treatment improved VCF optical signal in Anap labeled *Ci*-VSP and P2X2 receptor

To overcome the problems of small fluorescence changes and low incidence of successful detection of fluorescence changes, a small molecule kinase inhibitor, namely a Salt-inducible Kinase (SIK) Inhibitor (HG-9-91-01), was applied by injection into the oocytes, to decrease the intrinsic background fluorescence (*Lee and Bezanilla, 2019*). This inhibitor promotes UV-independent skin pigmentation, by increasing the production of melanin (*Mujahid et al., 2017*), resulting in a darker surface of the animal pole of the oocyte. As the intrinsic background fluorescence of the oocytes is decreased, the percentage of fluorescence change (ΔF/F) is expected to increase.

Optimization of SIK inhibitor treatment in VCF experiments using Anap as fluorophore was achieved for the following conditions: (1) the concentration of SIK inhibitor giving the maximum decrease in intrinsic background fluorescence; (2) the location of the microinjection (nuclear or cytoplasmic) and the duration of incubation.

Voltage-sensing phosphatase (*Ci*-VSP) F401Anap (*Sakata et al., 2016*) was used as a positive control to obtain reproducible and distinct results (*Figure 2A-E*). Oocytes were pre-treated with two concentrations of SIK inhibitor (30 nM and 300 nM, reflecting the concentration of injected solution). 300 nM SIK application increased ΔF/F more than twice that of non-treated oocytes, whereas the application of 30 nM did not give a significant increase (ΔF/F = 10.6% ± 2.5 at 500 nm [n = 6]; ΔF/

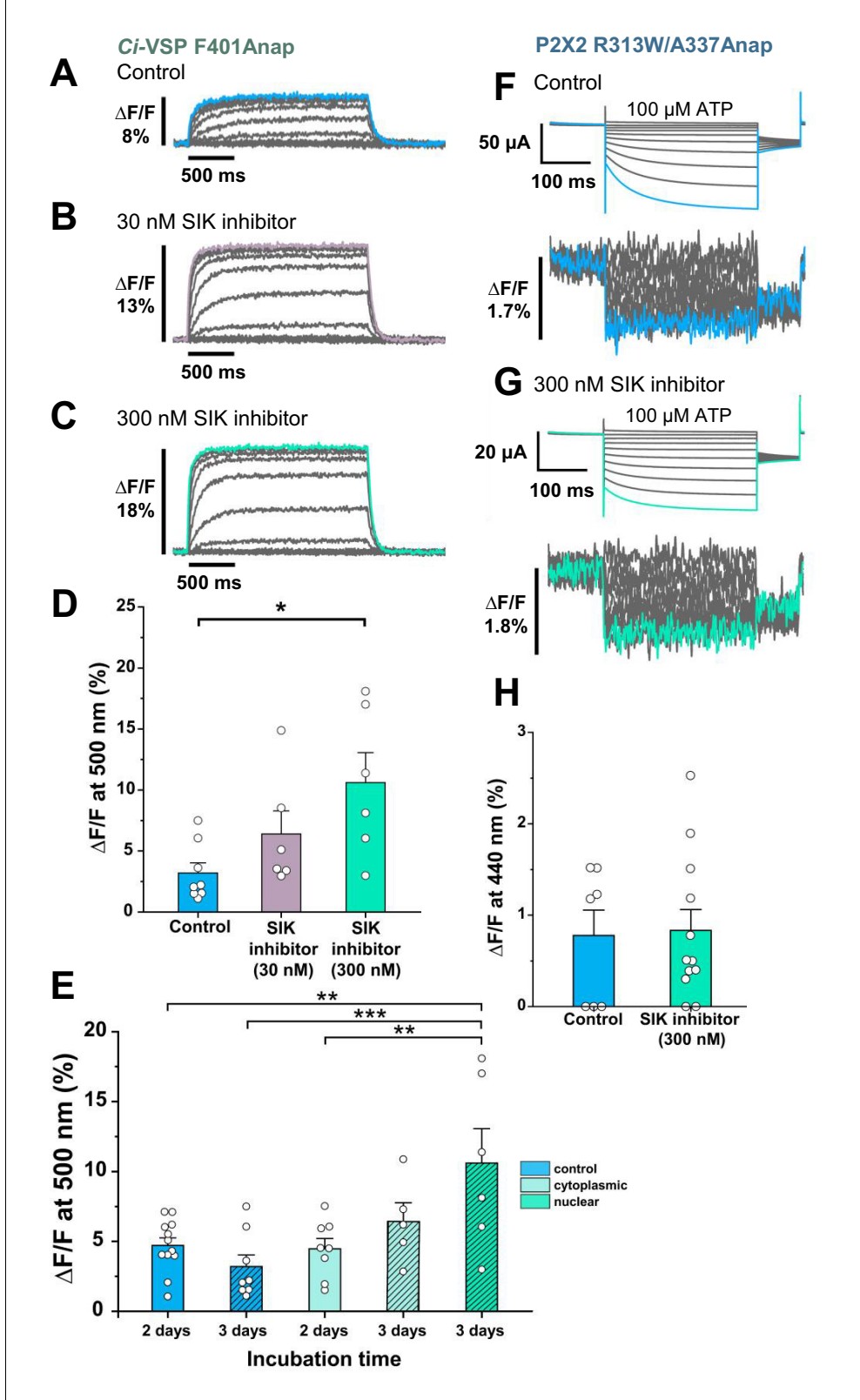

**Figure 2.** Effect of SIK inhibitor treatment in Anap-incorporated *Ci*-VSP and P2X2 receptor. SIK inhibitor treatment improved the VCF optical signal. (**A–C**) Representative fluorescence signal of VCF recordings of *Ci*-VSP without SIK inhibitor treatment, with 30 nM, and with 300 nM SIK inhibitor treatment ($\Delta F/F = 3.2\% \pm 0.8$ at 500 nm [n = 8]; $\Delta F/F = 6.4\% \pm 1.9$ [n = 6]; and $\Delta F/F = 10.6\% \pm 2.5$ [n = 6], respectively). (**D**) Comparison of non-treated (control) group

*Figure 2 continued on next page*

*Figure 2 continued*

(n = 8), 30 nM (n = 6), and 300 nM SIK inhibitor application (n = 6); *$p \leq 0.05$, p=0.01639, one-way ANOVA with Tukey's post-hoc test for 300 nM, compared to the control group. (E) Comparison of the incubation time and site of injection of SIK inhibitor treatment using 300 nM SIK inhibitor: control group, 2 days incubation (n = 12), control group, 3 days incubation (n = 8), SIK inhibitor treatment with cytoplasmic injection with 2 days incubation (n = 8), with cytoplasmic injection for 3 days (n = 5), with nuclear injection for 3 days (n = 6); **$p \leq 0.01$, ***$p \leq 0.001$, one-way ANOVA with Tukey's post-hoc test. (F, G) Representative current traces and fluorescence signal of VCF recordings of P2X2 receptor (A337Anap/R313W) without SIK inhibitor treatment and with the application of 300 nM SIK inhibitor ($\Delta F/F$ = 0.77% ± 0.3 at 440 nm [n = 7]; and $\Delta F/F$ = 0.83% ± 0.2 [n = 12], respectively). (H) A comparison of non-treated (control) group (n = 7) and 300 nM SIK inhibitor application (n = 12) (p=0.88107, two-sample t-test for 300 nM compared to the control group). All error bars are ± s.e.m. centered on the mean. Source data are provided in *Figure 2—source data 1*. Statistical analysis data are provided in *Figure 2—source data 2*, *Figure 2—source data 3*, and *Figure 2—source data 4*.

The online version of this article includes the following source data and figure supplement(s) for figure 2:

**Source data 1.** Effect of SIK inhibitor treatment in Anap-incorporated *Ci*-VSP and P2X2 receptor.
**Source data 2.** Statistical analysis to support graph in *Figure 2D*.
**Source data 3.** Statistical analysis to support graph in *Figure 2E*.
**Source data 4.** Statistical analysis to support graph in *Figure 2H*.
**Figure supplement 1.** Effect of 300 nM SIK inhibitor application on the incidence of detectable Anap fluorescence signal change of P2X2 receptor.
**Figure supplement 1—source data 1.** Effect of 300 nM SIK inhibitor application on the incidence of detectable Anap fluorescence signal change of P2X2 receptor.

---

F = 3.2% ± 0.8 [n = 8]; and $\Delta F/F$ = 6.4% ± 1.9 [n = 6], respectively, *Figure 2A-D*). This showed that 300 nM SIK inhibitor injected into the oocytes could decrease the intrinsic background fluorescence of the oocytes, thus increasing $\Delta F/F$.

Subsequently, a second series of optimization experiments was performed. In all of the following experiments, 300 nM SIK inhibitor was used. Control groups consisted of non-treated oocytes, which were incubated for either 2 or 3 days, resulting in $\Delta F/F$ = 4.7% ± 0.5 (n = 12) and $\Delta F/F$ = 3.2% ± 0.8 (n = 8), respectively.

The nuclear injection group, which was incubated for 3 days, had a larger $\Delta F/F$ than the other groups ($\Delta F/F$ = 10.6% ± 2.5 at 500 nm; n = 6). The cytoplasmic injection groups, which were incubated for either 2 or 3 days, resulted in $\Delta F/F$ = 4.5% ± 0.7 (n = 8) and $\Delta F/F$ = 6.4% ± 1.3 (n = 5) respectively. These results suggest that the optimal conditions for SIK inhibitor treatment are nuclear injection with 300 nM SIK inhibitor and 3 days incubation (*Figure 2E*).

After the optimal concentration, injection method, and incubation period were determined for the *Ci*-VSP experiment, the SIK inhibitor was then applied to the P2X2 A337Anap/R313W mutant (*Figure 2F,G*). R313W is a mutation which decreases the basal current in the absence of ATP, and the details are described later in Figure 4 and Figure 4—figure supplement 1. 300 nM SIK inhibitor treatment did not make any significant difference, in terms of the percentage of the fluorescence change compared to the control group ($\Delta F/F$ = 0.77% ± 0.3 at 440 nm [n = 7] and $\Delta F/F$ = 0.83% ± 0.2 at 440 nm [n = 12], respectively, *Figure 2H*). However, in the analysis of the incidence of detectable $\Delta F$ of Anap, the group treated with 300 nM SIK inhibitor showed a higher incidence than the control group (control = 57% [n = 7]; 300 nM SIK inhibitor application = 80% [n = 12]; *Figure 2—figure supplement 1A, B*). These results showed that in the case of P2X2, SIK inhibitor treatment improved the incidence of detectable $\Delta F/F$. Therefore, we decided to use the SIK inhibitor in all of the following experiments.

## ATP- and voltage-evoked Anap fluorescence changes at A337 and I341 in TM2 exhibit fast kinetics and linear voltage dependence

Application of 300 nM SIK inhibitor increased the incidence of $\Delta F$ (68.8 ± 3.2% [n = 6–9] with inhibitor, vs. 14.3 ± 4.1% [n = 5–16] without inhibitor) in A337Anap, with an improved signal-to-noise ratio ($\Delta F/F$ = 1.5% ± 0.2 at 440 nm [n = 8], *Figure 3A*). VCF recordings were performed by the application of 10 µM ATP and voltage step pulses from +40 mV to −140 mV, with a holding potential of +20 mV. Fluorescence intensity change occurred almost instantaneously, in less than 5 ms (*Figure 3B*). This showed that the kinetics of $\Delta F/F$ are very rapid and faster than the time course of voltage-

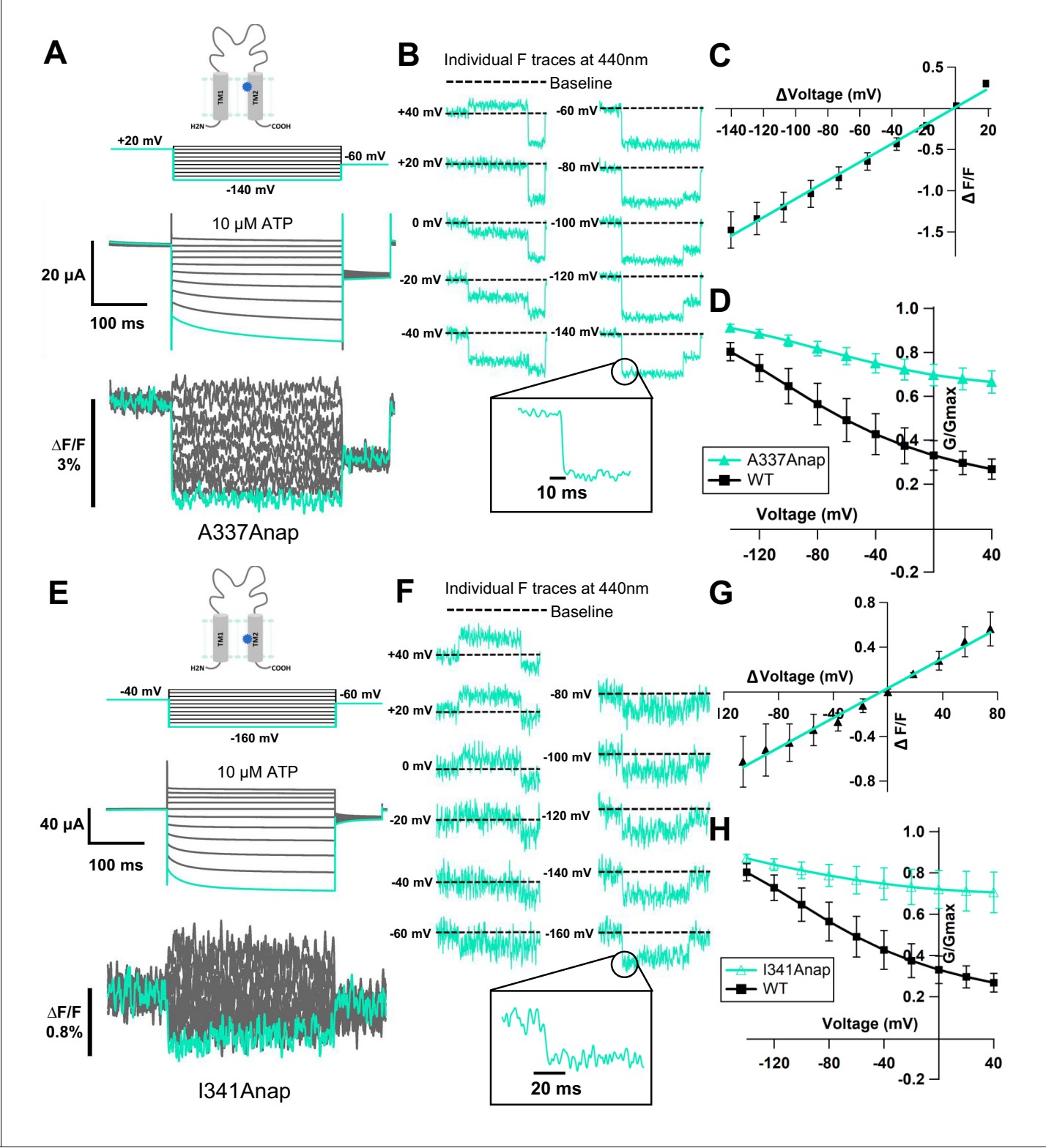

**Figure 3.** Voltage-clamp fluorometry (VCF) of Anap-incorporated P2X2 receptor in the presence of 300 nM SIK inhibitor upon ATP and voltage stimuli. The focused electric field converged at A337 and I341 in TM2, throughout P2X2 ATP- and voltage-dependent gating. (A) Representative current traces and fluorescence signal of VCF recordings at A337, with 300 nM SIK inhibitor treatment, in the presence of 10 μM ATP (ΔF/F = 1.5% ± 0.2 at 440 nm; n = 8). (B) Individual fluorescence traces during each voltage step at 440 nm. Inset shows fluorescence changes exhibiting fast kinetics in ms range. (C) F-V relationship showed a linear voltage dependence. Each X-axis for F-V relationship is ΔV from the holding potential. (D) G-V relationship comparison
*Figure 3 continued on next page*

*Figure 3 continued*

between A337Anap (turquoise filled triangle) and wildtype (black filled square) for 10 µM ATP (n = 8). Normalization was done based on the maximum conductance in the same concentration of ATP (10 µM) for each construct. (E) Representative current traces and fluorescence signal of VCF recordings at I341, with 300 nM SIK inhibitor treatment, in the presence of 10 µM ATP (ΔF/F = 0.6% ± 0.2 at 440 nm [n = 3]). (F) Individual fluorescence traces in each voltage step at 440 nm. Inset shows fluorescence changes also exhibiting fast kinetics in ms range. (G) F-V relationship showed a linear voltage dependence. Each X-axis for F-V relationship is ΔV from the holding potential. (H) G-V relationship comparison between I341Anap (turquoise open triangle) and wildtype (black filled square) for 10 µM ATP (n = 3). Normalization was done based on the maximum conductance in the same concentration of ATP (10 µM) for each construct. All error bars are ± s.e.m. centered on the mean. Source data are provided in *Figure 3—source data 1*.

The online version of this article includes the following source data and figure supplement(s) for figure 3:

**Source data 1.** VCF of A337Anap and I341Anap in the presence of 300 nM SIK inhibitor upon ATP and voltage stimuli.

**Figure supplement 1.** Voltage-clamp fluorometry (VCF) of L334Anap and L338Anap in the presence of 300 nM SIK inhibitor, with ATP and voltage steps.

**Figure supplement 1—source data 1.** VCF of L334Anap and L338Anap in the presence of 300 nM SIK inhibitor, with ATP and voltage steps.

dependent current activation. This also correlates well with the speed of the actual membrane potential change achieved by voltage clamp. Furthermore, the ΔF/F–V relationship of A337Anap showed a linear voltage dependence (y = 0.011x + 0.016; $R^2$ = 0.99 (n = 8), *Figure 3C*) in the voltage range used. These analyses of fluorescence changes at A337 indicated that the downward fluorescence change is not associated with protein conformational change. Rather, it is more likely related to an electrochromic effect.

Electrochromic effect is a shift in the fluorophore emission spectrum due to the interaction between two components: the fluorophore electronic state and the local electric field (*Bublitz and Boxer, 1997*; *Klymchenko and Demchenko, 2002*; *Dekel et al., 2012*). It has two distinctive characteristics: (1) fast kinetics of fluorescent change ($ΔF_{Fast}$); (2) linear voltage dependence of the F-V relationship (*Asamoah et al., 2003*; *Klymchenko et al., 2006*). The electrochromic effect in some voltage-sensitive dyes is used to directly detect the change of membrane potential by attaching the dye to the cell membrane. If the fluorophore is directly attached in a site-specific manner within ion channels/receptors as shown by studies in the *Shaker* B K$^+$ channel (*Asamoah et al., 2003*) and M$_2$ muscarinic receptor (*Dekel et al., 2012*), the detection of electrochromic effect implies that there is a convergence of the electric field at the position where the fluorophore is attached. Thus, the observed fluorescence change at the position of A337 in the P2X2 receptor was explained to be due to the electrochromic effect, indicating that there is a focused electric field at A337 in the TM2 domain.

We noted that the G-V relationship for this mutant showed that a large fraction of the channel is already open, even at depolarized potentials, in 10 µM ATP, compared to wildtype (*Figure 3D*), because of the high density of the expressed channel, shown by a rather large current amplitude (>20 µA). A previous study showed that P2X2 channel properties are correlated with expression density (*Fujiwara and Kubo, 2004*). In the case of lower expression levels, A337Anap showed a wildtype-like phenotype. For the purpose of VCF experiments, however, a high expression level is needed to observe a detectable fluorescence change, and thus we needed to use oocytes with high expression, resulting in a lesser fraction of voltage-dependent activation. Nonetheless, we could still observe a weak voltage-dependent relaxation during hyperpolarization, and thus this fluorescence change still reflects an event occurring at or around the position of A337 when the receptor senses the change in membrane voltage.

Similarly, the application of 300 nM SIK inhibitor to I341Anap resulted in a clearer (ΔF/F = 0.6% ± 0.2 at 440 nm [n = 3], *Figure 3E*) and more frequent Anap ΔF/F (38.1 ± 9.2%; n = 5–9), compared to that without SIK inhibitor application (16.02 ± 0.6%; n = 6–13) upon voltage step application in 10 µM ATP. The fluorescence intensity changes also occurred almost instantaneously, in less than 5 ms (*Figure 3F*). The ΔF/F–V relationship of I341Anap upon voltage step pulses in the presence of 10 µM of ATP, from +40 mV to −160 mV with a holding potential at −40 mV, also showed a linear voltage dependence (y = 0.007x + 0.03; $R^2$ = 0.99; n = 3, *Figure 3G*).

Thus, ΔF observed at the position of I341 in the TM2 domain also did not correlate with hyperpolarization-induced conformational change. The changes are likely to be due to a phenomenon similar to that observed at the position of A337, which is related to the electrochromic effect. The G-V

relationship of this mutant in the presence of 10 µM ATP was not different from that of A337Anap, as shown in *Figure 3H*. Taking these results together, the observed fluorescence intensity changes at I341 and A337 in the TM2 domain are best explained by an electric field convergence close to both positions, which could be critical for the complex gating of the P2X2 receptor.

Since the Anap mutant scanning experiments were initially performed in the absence of an SIK inhibitor, there was a possibility that there were fluorescence changes which went undetected. Thus, we performed VCF recordings with SIK inhibitor treatment in some Anap mutants which had been screened before. The focus was on residues surrounding A337 and I341 in the TM1 and TM2 domains. In the re-screening experiments, we observed small ΔF at L334 and L338: L334Anap: ΔF/F = 0.38% ± 0.2 at 440 nm (n = 2), and L338Anap: ΔF/F = 0.26% ± 0.03 at 440 nm (n = 4), as shown in *Figure 3—figure supplement 1*. Moreover, the incidence of fluorescence change detection for L338Anap in three batches (20 cells in total) was 2/5, 0/5, and 2/10. For L334Anap it was 0/5, 0/10, and 2/6 in three batches (21 cells in total). These results suggest that the focused electric field might lie at TM2 from L334 down to I341, and that it is more strongly converged at A337 and I341, because the electrochromic signal at A337 and I341 was observed more frequently.

## Fluorescence change of Anap at A337 upon voltage change was also observed in the absence of ATP and was [ATP]-dependent

To ensure that the fluorescence changes observed at A337 upon voltage change were not due to a change of ion flux, as in the case of the K2P K⁺ channel (*Schewe et al., 2016*), recording was performed in the absence of ATP. In the same cells, VCF recordings were performed by applying voltage steps in the absence of ATP and then in the presence of 10 µM ATP.

When the voltage steps were applied in the absence of ATP, fluorescence changes were observed (ΔF/F = 1.9% ± 0.4 at 440 nm, n = 4, *Figure 4A-C*). The changes exhibited fast kinetics and a linear voltage dependence. ΔF/F in the absence of ATP was larger than that in the presence of 10 µM ATP (ΔF/F = 0.7% ± 0.1 at 440 nm [n = 4] *Figure 4A-C*). However, the A337Anap mutant showed a high basal activity, even in the absence of ATP, when the expression level was high. As observed in the current traces in no ATP, some of the channels expressed were already open (*Figure 4A*). Thus, the ΔF in 0 ATP observed in the above experiments might just represent the ΔF in the open state. To record the ΔF in the closed state with little current in no ATP, an additional mutation was introduced, which suppresses the basal activity by stabilizing the closed state.

The extracellular linker plays important roles in transmitting the signal from the ATP binding pocket (ECD domain) to the TM domains (*Keceli and Kubo, 2014*). It includes β−14, which directly links the ATP binding site with the TM2 domain. Mutation of the β−14 residue R313 to phenylalanine or tryptophan stabilized the closed state of the P2X2 receptor, as seen in the G-V relationship in 100 µM ATP (*Figure 4—figure supplement 1A–D*). This mutation was introduced into A337Anap (A337Anap/R313F or A337Anap/R313W) to determine whether the ΔF related to the focused electric field is present at A337, even when the channel is mostly closed in 0 ATP.

Results from VCF recording of both A337Anap/R313F (*Figure 4D-I*) and A337Anap/R313W (*Figure 4—figure supplement 1F–K*) confirmed that the focused electric field is present at A337 even when the channel is mostly closed. VCF recording in the absence of ATP for A337Anap/R313F showed a remarkable ΔF/F with mostly closed channels when voltage step pulses were applied (ΔF/F = 2.6% ± 0.3 at 440 nm; n = 8; *Figure 4D-F*). 30 µM ATP application resulted in smaller ΔF/F than in 0 ATP (ΔF/F = 1.7% ± 0.2 at 440 nm; n = 8; *Figure 4D-F*). These results confirmed that ΔF/F at the position of A337 is larger in the absence of ATP than in the presence of ATP. It was of interest to know whether or not the concentration of ATP affects ΔF/F at A337. Therefore, a higher concentration of ATP (100 µM) was used in a similar series of experiments. Fluorescence changes were again larger in the absence of ATP (ΔF/F = 2.4% ± 0.3 at 440 nm; n = 8; *Figure 4G-I*) than in the presence of 100 µM ATP (ΔF/F = 0.9% ± 0.1; n = 8; *Figure 4G-I*). ΔF/F was shown to become smaller with an increase in [ATP], by comparing ΔF/F in the presence of 30 µM and 100 µM ATP. Similar series of experiments were also performed using the A337Anap/R313W construct (*Figure 4—figure supplement 1F–K*), and similar phenotypes were observed.

Taken together, these results show that the ΔF/F at A337 is [ATP]-dependent and larger in the absence of ATP. The higher ΔF/F implies a stronger focused electric field in 0 ATP, but it is also possible that the difference comes from the difference in the environment of the fluorophore, as

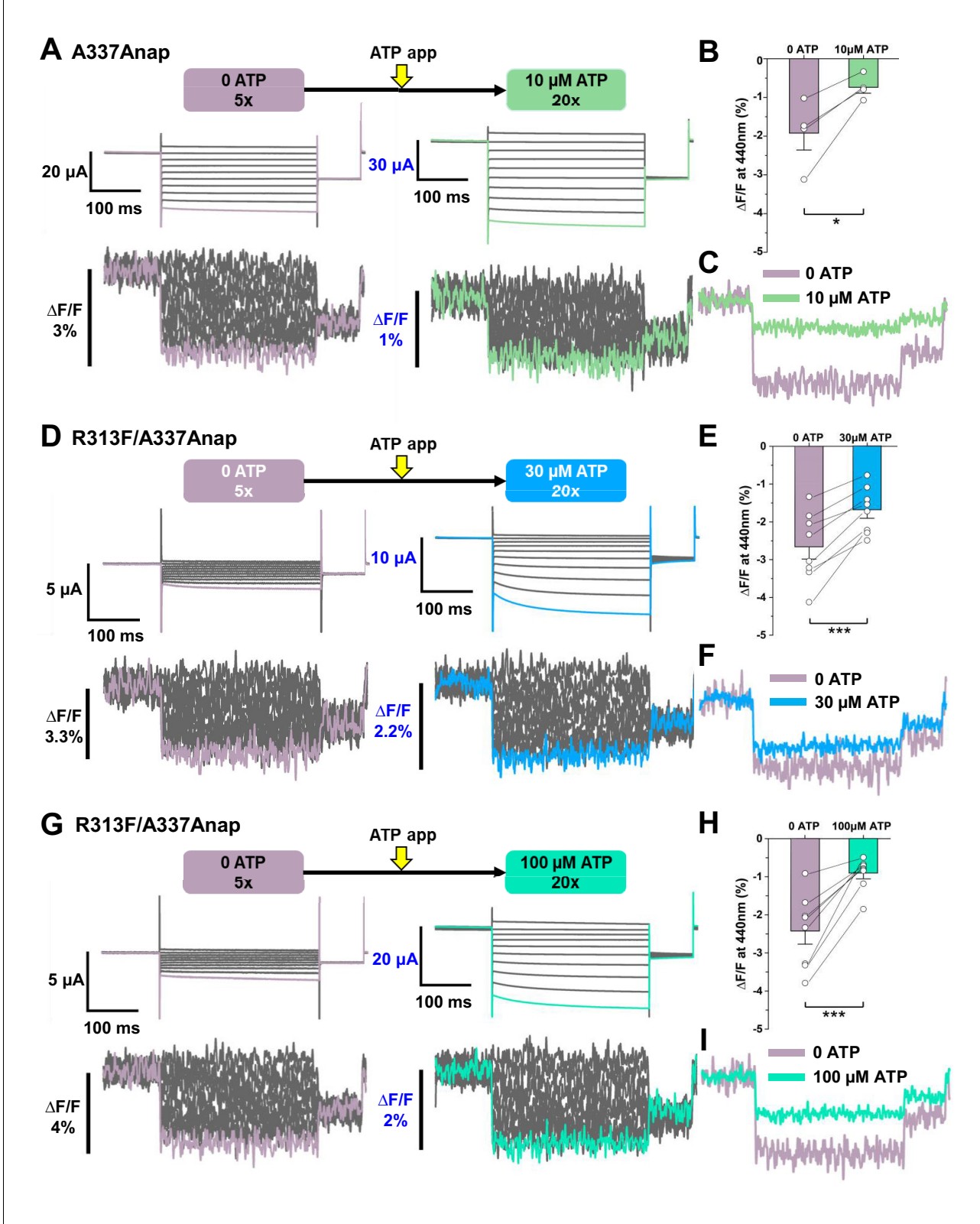

**Figure 4.** Voltage-clamp fluorometry (VCF) of Anap-labeled P2X2 at A337 in TM2 evoked by hyperpolarization in the absence and presence of ATP. Anap fluorescence changes at A337 were observed even in the absence of ATP upon hyperpolarization. (A) Representative current traces and fluorescence signal of VCF recordings at A337 in the absence of ATP (ΔF/F = 1.9% ± 0.4 at 440 nm (n = 4)) and in the presence of 10 μM ATP (ΔF/F = 0.7% ± 0.1 at 440 nm (n = 4)), from the same cell. (B) Comparison of the fluorescence changes in the absence and in the presence of 10 μM ATP

*Figure 4 continued on next page*

*Figure 4 continued*

(*p≤0.05, p=0.02876, paired t-test; n = 4). (**C**) Superimposed fluorescence traces at −140 mV, in 0 ATP (light purple) and 10 µM ATP (light green), from the same cell. (**D–I**) An additional R313F mutation was introduced to lower the basal activity of A337Anap and stabilize the closed state. (**D**) Representative current traces and fluorescence signal of VCF recordings of A337/R313F in the absence of ATP (ΔF/F = 2.6% ± 0.3 at 440 nm; n = 8) and in the presence of 30 µM ATP (ΔF/F = 1.7% ± 0.2; n = 8) from the same cell. (**E**) Comparison of the fluorescence changes in the absence and in the presence of 30 µM ATP (***p≤0.001, p=0.00045, paired t-test; n = 8). (**F**) Superimposed fluorescence traces at −140 mV, in 0 ATP (light purple) and 30 µM ATP (blue), from the same cell. (**G**) Representative current traces and fluorescence signal of VCF recordings of A337/R313F in the absence of ATP (ΔF/F = 2.4% ± 0.3 at 440 nm; n = 8) and in the presence of 100 µM ATP (ΔF/F = 0.9% ± 0.1; n = 8). (**H**) Comparison of the fluorescence changes in the absence and in the presence of 100 µM ATP (***p≤0.001, p=0.0005, paired t-test; n = 8). (**I**) Superimposed fluorescence traces at −140 mV in 0 ATP (light purple) and 100 µM ATP (turquoise), from the same cell. All error bars are ± s.e.m. centered on the mean. Source data are provided in *Figure 4— source data 1*. Statistical analysis data are provided in *Figure 4—source data 2*, *Figure 4—source data 3*, and *Figure 4—source data 4*.

The online version of this article includes the following source data and figure supplement(s) for figure 4:

**Source data 1.** VCF of Anap-labeled P2X2 at A337 evoked by hyperpolarization in the absence and presence of ATP.

**Source data 2.** Statistical analysis to support graph in *Figure 4B*.

**Source data 3.** Statistical analysis to support graph in *Figure 4E*.

**Source data 4.** Statistical analysis to support graph in *Figure 4H*.

**Figure supplement 1.** VCF of Anap-labeled P2X2 at A337 in TM2 evoked by hyperpolarization in the absence and presence of ATP.

**Figure supplement 1—source data 1.** VCF of Anap-labeled P2X2 at A337 evoked by hyperpolarization in the absence and presence of ATP.

**Figure supplement 1—source data 2.** Statistical analysis to support graph in *Figure 4—figure supplement 1G*.

**Figure supplement 1—source data 3.** Statistical analysis to support graph in *Figure 4—figure supplement 1J*.

discussed later (*Figure 6—figure supplement 1*). In any case, it was shown that a ΔF/F and thus a focused electric field are present both in the absence and the presence of ATP.

## Hyperpolarization-induced structural rearrangements were detected at and around A337 in TM2, upon the additional mutation of K308R

Upon ATP binding, the P2X receptor undergoes major structural rearrangements, which result in transitions from closed to open state, with remarkable alterations in three regions: the ATP binding site, the extracellular linker, which links ECD to TM domains, and the TM domains (*Kawate et al., 2009*; *Hattori and Gouaux, 2012*; *Mansoor et al., 2016*). There is a possibility that the P2X2 receptor could undergo relatively minor but important structural rearrangements in response to hyperpolarization of the membrane voltage, after the major structural rearrangements caused by the binding of ATP. A fraction of a slow ΔF and non-linear ΔF/F–V has not been detected so far. This might be due to suppression of voltage-dependent activation in high-expression oocytes, in which there is significant activity even at depolarized potentials (e.g., *Figure 3D,H*). Thus, an additional mutation which shows remarkable voltage-dependent activation, even in high expression conditions, is needed.

We then tested this possibility by introducing a K308R mutation into A337Anap. This charge-maintaining mutation, K308R, is shown to make the voltage-dependent activation more prominent, that is, it is least active at depolarized potentials, even in high-expression oocytes, and it also accelerates the activation kinetics of P2X2 upon voltage steps (*Keceli and Kubo, 2009*). K308 is a conserved residue located in the ATP binding site. It was shown to be important, not only for ATP binding (*Ennion et al., 2000*; *Jiang et al., 2000*; *Roberts et al., 2006*), but also for the conformational change associated with channel opening (*Cao et al., 2007*). If the voltage-dependent activation is more prominent, even in the high-expression cells for VCF experiments, there was a possibility that we might be able to detect ΔF associated with voltage-dependent gating.

VCF recording of K308R/A337Anap was performed in the presence of 300 µM ATP, while a voltage-step from +40 mV to −160 mV, with a holding potential of +20 mV, was applied. A high concentration of ATP was applied because K308R/A337Anap has a lower sensitivity to ATP. ATP-evoked current of K308/A337Anap was inhibited by both Suramin and PPADS, the P2X2 non-specific blockers, confirming that the current is indeed a P2X2 receptor current (*Figure 5—figure supplement 1*).

Hyperpolarization elicited fluorescence signals which consisted of two components, a very fast decrease ($\Delta F_{Fast}/F$) and a slow increase ($\Delta F_{Slow}/F$) to steady state ($\Delta F_{Steady\text{-}state}/F$) (*Figure 5A,B*). Plots of the F-V relationship at the end of the recording time interval (at the steady state) showed that F-V consisted of mixed components, a linear component and a non-linear component (*Figure 5C*). The

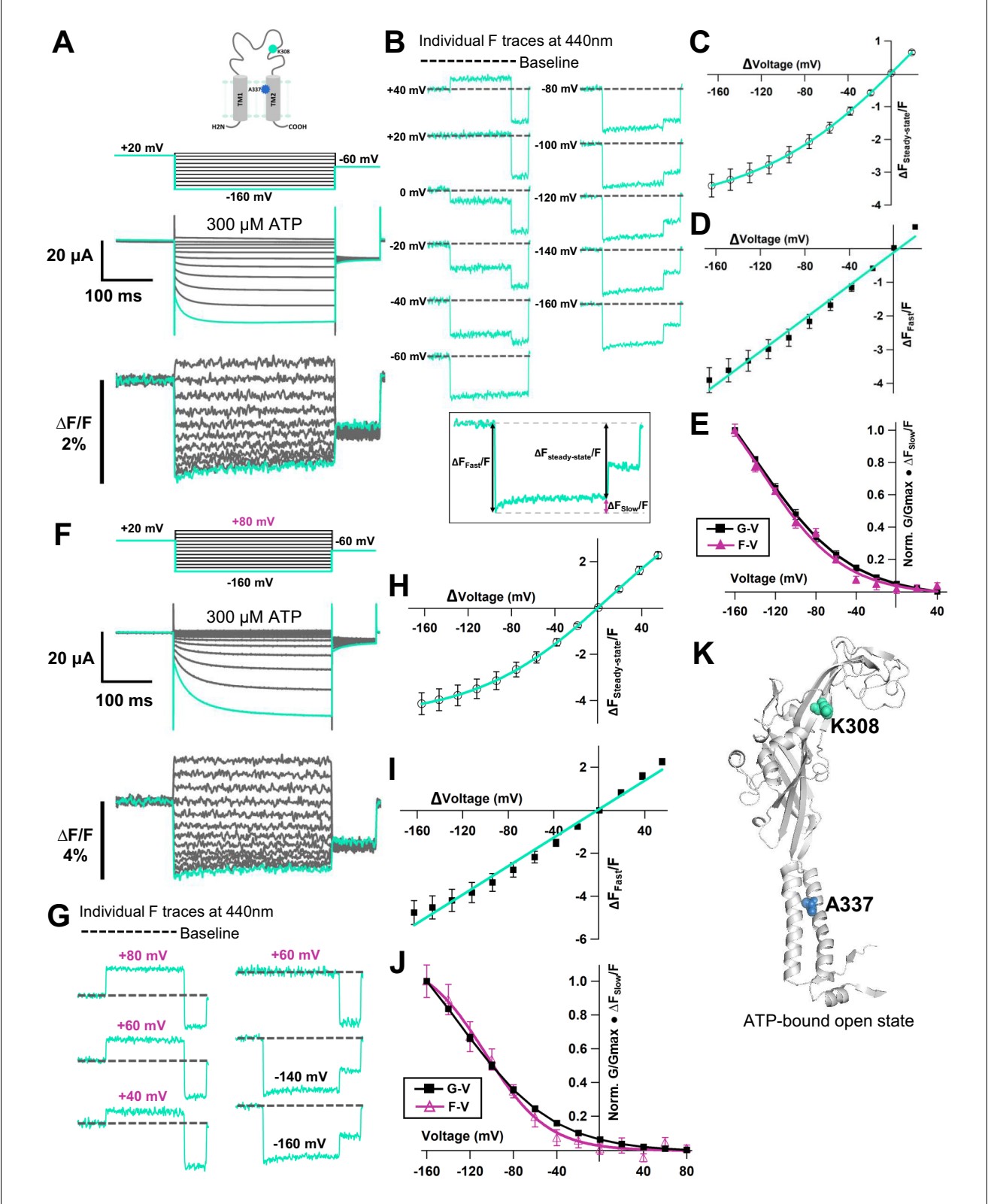

**Figure 5.** Voltage-clamp fluorometry (VCF) of Anap-labeled P2X2 at A337 in TM2 with the additional mutation of K308R evoked by hyperpolarization in the presence of ATP. (**A**) Representative current traces and fluorescence signal of VCF recordings of K308R/A337Anap with 300 nM SIK inhibitor treatment in the presence of 300 µM ATP, from +40 mV to −160 mV with a holding potential of +20 mV ($\Delta F_{Steady-state}/F$ = 3.4% ± 0.3 at 440 nm, n = 8). (**B**) Individual fluorescence traces at each voltage step. Inset shows that the fluorescence signal of K308R/A337Anap consists of two components,

*Figure 5 continued on next page*

Figure 5 continued

instantaneous downward change ($\Delta F_{Fast}$/F) and slow upward change ($\Delta F_{Slow}$/F). (C) F-V relationship of the mixed component ($\Delta F_{Steady-state}$/F) was calculated from the last 50 ms of fluorescence signal. Component of $\Delta F_{Steady-state}$/F is shown in inset of (B). $F_{Steady-state}$-V relationship shows that it consists of only a linear component at depolarized potentials, and there are mixed components at hyperpolarized potentials. (D) $F_{Fast}$-V relationship was taken from the first 5 ms of the fluorescence signal. $F_{Fast}$-V relationship showed almost linear voltage dependence ($\Delta F_{Fast}$/F = 3.9% ± 0.4 at 440 nm; n = 8). (E) Comparison of $F_{Slow}$-V and G-V relationships. Purple filled triangle trace shows $F_{Slow}$-V relationship extracted from the fluorescence traces depicted in inset (B), as shown by purple arrow, from the equation $\Delta F_{steady-state}$/F = $\Delta F_{fast}$/F + $\Delta F_{slow}$/F. Normalization was based on the maximum $\Delta F_{slow}$/F (at −160 mV). Black filled square trace shows G-V relationship in the presence of 300 μM ATP. Normalization was based on the maximum conductance in the same concentration of ATP (300 μM). (F) Representative current traces and fluorescence signal of VCF recordings of K308R/A337Anap with 300 nM SIK inhibitor treatment, in the presence of 300 μM ATP, at more depolarized potentials from up to +80 mV to −160 mV, with a holding potential of +20 mV ($\Delta F_{Steady-state}$/F = 4.1% ± 0.5 at 440 nm; n = 5). (G) Individual fluorescence traces at each depolarized voltage step and some hyperpolarized voltage steps. (H) $F_{Steady-state}$-V relationship further confirms that it consists of a linear component and a slow component generated only upon hyperpolarization. (I) $F_{Fast}$-V relationship shows almost linear voltage dependence ($\Delta F_{Fast}$/F = 4.7% ± 0.5 at 440 nm; n = 5). (J) Comparison of $F_{Slow}$-V and G-V relationships. Purple open triangle trace shows $F_{Slow}$-V relationship extracted from the fluorescence traces depicted in (F). Normalization was based on the maximum $\Delta F_{slow}$/F (at −160 mV). Black filled square trace shows G-V relationship in the presence of 300 μM ATP. Normalization was based on the maximum conductance in the same concentration of ATP (300 μM). All error bars are ± s.e.m. centered on the mean. (K) Side view structure of the position of K308 and A337 in the ATP-bound open state. Source data are provided in *Figure 5—source data 1*.

The online version of this article includes the following source data and figure supplement(s) for figure 5:

**Source data 1.** VCF of K308R/A337Anap evoked by hyperpolarization in the presence of ATP.
**Figure supplement 1.** ATP-evoked currents in K308R/A337Anap were inhibited by P2X2 non-specific blockers: Suramin and PPADS.
**Figure supplement 1—source data 1.** ATP-evoked currents of K308R/A337Anap were inhibited by Suramin and PPADS.
**Figure supplement 2.** Slow fraction of fluorescence changes at A337 in TM2 with the additional mutation of K308R, evoked by hyperpolarization, was [ATP]-dependent.
**Figure supplement 2—source data 1.** Slow fraction of fluorescence changes at K308R/A337Anap, evoked by hyperpolarization, was [ATP] dependent.
**Figure supplement 2—source data 2.** Statistical analysis to support the graph in *Figure 5—figure supplement 2G*.

presence of the two components suggests that they might result from two different mechanisms. The F-V relationship of $\Delta F_{Fast}$/F showed a linear voltage dependence, which is similar to the F-V for A337Anap alone, which was generated from the electrochromic signal (*Figure 3C*, *Figure 5D*). In contrast, the F-V relationship of $\Delta F_{Slow}$/F showed a non-linear voltage dependence. The F-V and G-V relationships of the slow component overlap very well (*Figure 5E*), showing that the slow $\Delta F$ reflects the hyperpolarization-induced structural rearrangements that occur at and around the position of A337.

It was of interest to see the correlation between [ATP] and the $\Delta F_{Slow}$/F in this study. We performed additional VCF recordings of K308R/A337Anap in the presence of 30 μM ATP (*Figure 5— figure supplement 2A*). In the presence of 30 μM ATP, hyperpolarization again elicited fluorescence signals which consist of two components, a very fast decrease ($\Delta F_{Fast}$/F) and a slow increase ($\Delta F_{Slow}$/F) to steady state ($\Delta F_{Steady-state}$/F) (*Figure 5—figure supplement 2A*). Plots of the F-V relationship at the end of the recording time interval (at the steady state) again showed that F-V consists of mixed components, a linear component, and a non-linear component (*Figure 5—figure supplement 2B*). The F-V relationship of $\Delta F_{Fast}$/F again showed a linear voltage dependence (*Figure 5—figure supplement 2C*), and the F-V relationship of $\Delta F_{Slow}$/F showed a non-linear voltage dependence (*Figure 5—figure supplement 2D*), similar to the case in 300 μM ATP (*Figure 5*).

We were curious to know whether the time constant of $\Delta F_{Slow}$/F varied with [ATP]. We then analyzed the time constant of activation of the $\Delta F_{Slow}$/F in 30 μM ATP and 300 μM ATP. However, the analysis of the activation time constant of $\Delta F_{Slow}$ could not be performed reliably, due to a small $\Delta F_{Slow}$ component and noise which made it difficult to fit an exponential function. Nevertheless, the results of the analysis are shown in *Figure 5—figure supplement 2E–2G*. Current traces could be well fitted with a single exponential function (*Figure 5—figure supplement 2E–F*). However, as to the fluorescence traces, as shown in *Figure 5—figure supplement 2E–F*, $\Delta F_{Slow}$ could be fitted at −160 mV but not reliably at −140 mV or −120 mV. The analysis for −160 mV showed that the time constant of the voltage-dependent $\Delta F$ varied with [ATP]. The time constant of voltage-dependent $\Delta F$ was significantly larger in the lower [ATP], as depicted in *Figure 5—figure supplement 2G*.

Next, we examined whether $\Delta F_{Slow}$ is indeed generated only at hyperpolarized potentials, to obtain evidence of voltage-dependent structural rearrangements during P2X2 receptor complex gating. We performed VCF recordings by applying step pulses from +80 mV to −160 mV, with a

holding potential of +20 mV. The F-V relationship in the steady state showed a mixed signal. This set of recordings showed that at more depolarized potentials the fluorescence signal consists only of a linear component (*Figure 5F-H*). Separation of the mixed fluorescence signal also resulted in a rapidly changing linear F-V for $\Delta F_{Fast}/F$ (*Figure 5I*) and a non-linear F-V for $\Delta F_{Slow}/F$ (*Figure 5J*), with no slow component from +80 mV to 0 mV.

The results further confirm that the slow rise in K308R/A337Anap fluorescence signal reflects structural rearrangements at and around the position of A337 in response to changes in membrane voltage.

## Fluorescence signal changes at A337Anap/K308R exhibited only the fast component in the absence of ATP and showed two components in the presence of ATP

We also examined whether the non-linear component of the K308R/A337Anap fluorescence signal was abolished in the absence of ATP. We performed VCF recordings of the same cell, by applying voltage steps in the absence of ATP and in the presence of 300 µM ATP. In the absence of ATP, the fluorescence signal consisted of only one component, the fast component ($\Delta F_{Fast}/F$, *Figure 6A,B*). The F-V relationship for $\Delta F_{Fast}/F$ was linear and is thought to be derived from the electrochromic phenomenon, showing that A337 is located in the focused electric field (*Figure 6C*).

Subsequently, when the voltage step pulses were applied in the presence of 300 µM ATP, the slow component could be observed (*Figure 6A,B,D*). The F-V relationship in the steady state showed a mixture of the two components (*Figure 6D*). Separation of this mixed component resulted in a linear F-V for the fast component (*Figure 6E*) and a non-linear F-V for the slow component (*Figure 6F*), which is consistent with the previous experiments. Taken together, these results further show that the slow component of the fluorescence intensity changes reflects structural rearrangements of the P2X2 receptor, which depend on both [ATP] and voltage.

Additionally, results consistent with *Figure 4E,F,H, and I* were also obtained in terms of the fluorescence intensity change of the fast component. $\Delta F_{Fast}/F$ in the absence of ATP was larger than in the presence of ATP ($\Delta F_{Fast}/F$ = 4.4% ± 0.5 at 440 nm [n = 6] and $\Delta F_{Fast}/F$ = 1.7% ± 0.3 at 440 nm [n = 6]; *Figure 6B*). However, there was a concern that F itself significantly changed due to a difference in the environment surrounding A337Anap in the closed and open states. We performed experiments to address this by measuring the absolute F (output of the photomultiplier tube) in the absence and in the presence of ATP. Using the K308R/A337Anap construct, each recording was repeated five times, for averaging, in the same cell as shown in *Figure 6—figure supplement 1*. There was a slight reduction, not increase, in the F in the presence of ATP, which could be due to fluorescence bleaching (*Figure 6—figure supplement 1A,B*). Thus, a larger $\Delta F/F$ in the absence of ATP is thought not to be due to less F, and we concluded that $\Delta F/F$ was larger in the absence of ATP. The noise level is similar (*Figure 6—figure supplement 1C*), showing that the apparent larger noise level in 0 ATP (*Figure 4C, F, and I*; *Figure 4—figure supplement 1H,K*; *Figure 6B*) is not due to smaller F but due to a lower number of averaged traces.

## A337 in TM2 might interact with F44 in TM1 to stabilize the open state of the P2X2 receptor

The electric field convergence at A337 and I341 and the voltage-dependent conformational changes at or around A337 could provide us with a clue to understand the mechanism of the complex gating of the P2X2 receptor. The existence of a strong electric field supports the possible location of a key residue which is responsible for voltage sensing (*Asamoah et al., 2003*; *Dekel et al., 2012*). Thus, various single amino acid mutations were introduced at the position of A337, and their electrophysiological properties were analyzed, focusing on the [ATP]- and voltage-dependent gating properties, to see whether or not this amino acid plays an important role in the P2X2 complex gating (*Figure 7A,B*).

Mutations to A337R, A337K, and A337D had severe effects. When the voltage step pulses were applied in 30 µM ATP, these mutants almost lacked voltage sensitivity. A337E, A337Y, and A337F showed voltage sensitivity with various activation kinetics. The most striking changes were observed in A337Y and A337F. The activation evoked by a voltage step was clearly different from wildtype, whereas the A337E mutation had a less severe effect (*Figure 7A*). G-V relationships in 30 µM ATP

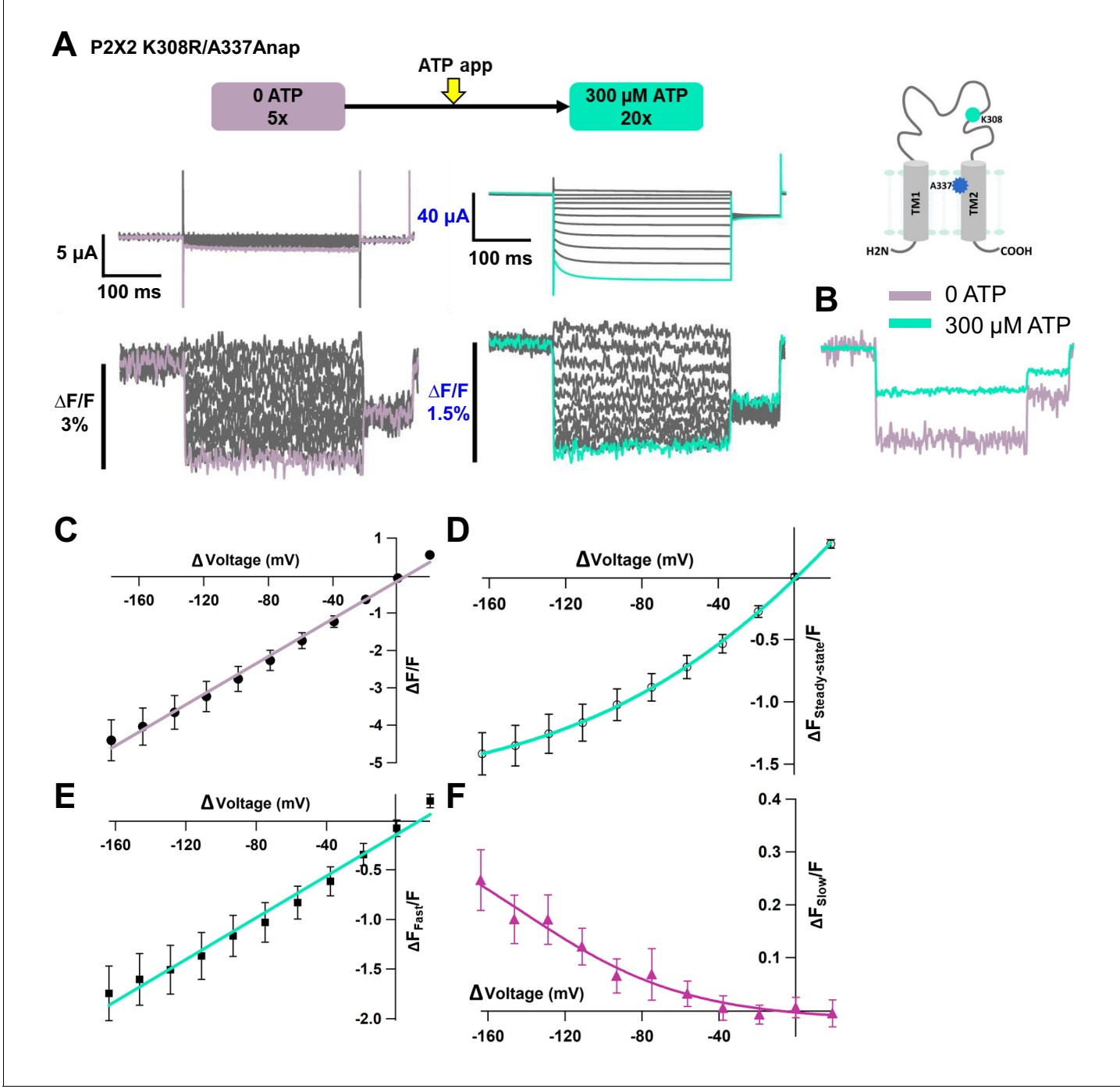

**Figure 6.** Voltage-clamp fluorometry (VCF) of Anap-labeled P2X2 at A337 in TM2 with the additional mutation of K308R, evoked by hyperpolarization in the absence and presence of ATP. Fluorescence signal changes at K308R/A337Anap exhibited only a fast component in the absence of ATP and consisted of two components in the presence of ATP. (A) Representative current traces and fluorescence signal of VCF recordings of K308R/A337Anap in the absence of ATP ($\Delta F/F = 4.4\% \pm 0.5$ at 440 nm; n = 6) and in the presence of 300 μM ATP ($\Delta F_{Steady-state}/F = 1.4\% \pm 0.2$; n = 6), from the same cell. (B) Superimposed fluorescence traces at −160 mV in 0 ATP (light purple) and 300 μM (turquoise). (C) F-V relationship, in the absence of ATP, taken from the last 100 ms of the fluorescence signals, shows a linear voltage dependence ($R^2 = 0.99$); therefore, it has only the fast component ($\Delta F_{Fast}/F$). (D) F-V relationship, in the presence of 300 μM ATP, taken from the last 50 ms ($\Delta F_{Steady-state}/F$) of the fluorescence signals shows mixed components. (E–F) F-V relationship from two separate components of the fluorescence signal change, in the presence of 300 μM ATP. (E) $F_{Fast}$-V relationship ($\Delta F_{Fast}/F = 1.7 \pm 0.3$ at 440 nm; n = 6) shows almost linear voltage dependence ($R^2 = 0.98$). (F) $F_{Slow}$-V relationship ($\Delta F_{Slow}/F = 0.25 \pm 0.05$ at 440 nm; n = 6). The X-axis for the F-V relationship is ΔV from the holding potential. All error bars are ± s.e.m. centered on the mean. Source data are provided in *Figure 6—source data 1*.

*Figure 6 continued on next page*

*Figure 6 continued*

The online version of this article includes the following source data and figure supplement(s) for figure 6:

**Source data 1.** VCF of K308R/A337Anap evoked by hyperpolarization in the absence and presence of ATP.

**Figure supplement 1.** Voltage-clamp fluorometry (VCF) of Anap-labeled P2X2 at A337 in TM2 with the additional mutation of K308R, evoked by hyperpolarization in the absence and presence of ATP.

**Figure supplement 1—source data 1.** VCF of K308R/A337Anap evoked by hyperpolarization in the absence and presence of ATP.

**Figure supplement 1—source data 2.** Statistical analysis to support graph in *Figure 6—figure supplement 1B*.

for mutants and wildtype were analyzed (*Figure 7B*). Normalization was based on the maximum conductance at the highest ATP concentration (300 µM) for each construct. Here we could also see that A337Y and A337F preferred to stay in the closed state. Thus, the alteration of activation kinetics and voltage dependence by mutation of A337 showed that this position is critical for the P2X2 receptor complex gating.

Next, we aimed to identify the counterpart in the TM1 domain with which A337 might have an interaction during complex gating. Based on the crystal structure data of *h*P2X3 in the closed and ATP-bound open states (PDB ID: 5SVJ, 5SVK, respectively) (*Mansoor et al., 2016*), homology modeling of *r*P2X2 showed that F44 in the TM1 domain rotates and moves towards A337 upon ATP binding (*Figure 7C,D*). Various single amino acid mutations were then introduced at F44, and the [ATP]- and voltage-dependent gating was analyzed (*Figure 7E,F*).

The F44A mutation strikingly changed the gating. It showed a relatively high basal current in the absence of ATP and further responded to ATP application. Voltage-dependent gating was also changed, as seen in the lack of tail current, showing that this mutant might have constitutive activity with rectified permeation properties. Mutation to positively charged residues (F44R, F44K) resulted in a non-functional channel and/or a very low expression level, as the recording on day four did not evoke any response to the highest concentration of ATP used in this study (300 µM). Mutation to negatively charged residues (F44E, F44D) and aromatic residues (F44Y, F44W) remarkably changed the ATP-evoked response (*Figure 7E*). All four mutants still opened upon the application of ATP, but current decay in the continuous presence of ATP appeared to be faster than wildtype.

F44 is conserved only in P2X2 and P2X3, within the P2X family. Other subtypes of P2X receptor, such as P2X1, P2X4, P2X6, and P2X7, except P2X5, have valine at the corresponding position (*Kawate et al., 2009*). Thus, the F44V mutation was also introduced. 10 µM of ATP could activate F44V but resulted in faster current decay than wildtype.

Voltage step pulses were applied during the course of current decay, because there was no clear steady state (*Figure 7E*). Nonetheless, the effect of the mutation at F44V on the voltage-dependent gating could still be observed. The G-V relationship of F44V in 10 µM ATP showed that this mutant was far less sensitive to voltage than wildtype (*Figure 7F*). Taken together, the results of the mutations introduced at position F44 showed that this residue is critical for the proper ATP- and voltage-dependent gating of the P2X2 receptor.

Additionally, as the single amino acid mutations at both A337 and F44 altered the gating of P2X2, it was of interest to determine whether the introduction of swapped mutations into A337/F44 would rescue the wildtype phenotype. The phenotype of F44A/A337F was similar to F44A, and the wildtype phenotype was not rescued (*Figure 7—figure supplement 1*). It is possible that an interaction between A337 and F44 could not be properly formed in the swapped mutant.

Next, an artificial electrostatic bridge was introduced between A337 and F44 to prove that the interaction between the two residues is critical in the ATP-bound open state. Various paired electrostatically charged residues were introduced into A337 and F44, in order to see if the artificial electrostatic bridge could be formed. The F44E/A337R pair showed constitutive activity. This double mutant was already open before ATP application and did not show any response to ATP (*Figure 7G*). When voltage step pulses were applied, this mutant lacked sensitivity to voltage, with a rectified permeation property, as seen by the total lack of tail currents (*Figure 7H*). Additionally, comparison of the current amplitude before and after ATP application showed that F44E/A337R was already open before ATP application (*Figure 7I*). The results showed that A337 in the TM2 domain might interact with F44 in TM1 to stabilize the open state of the P2X2 receptor.

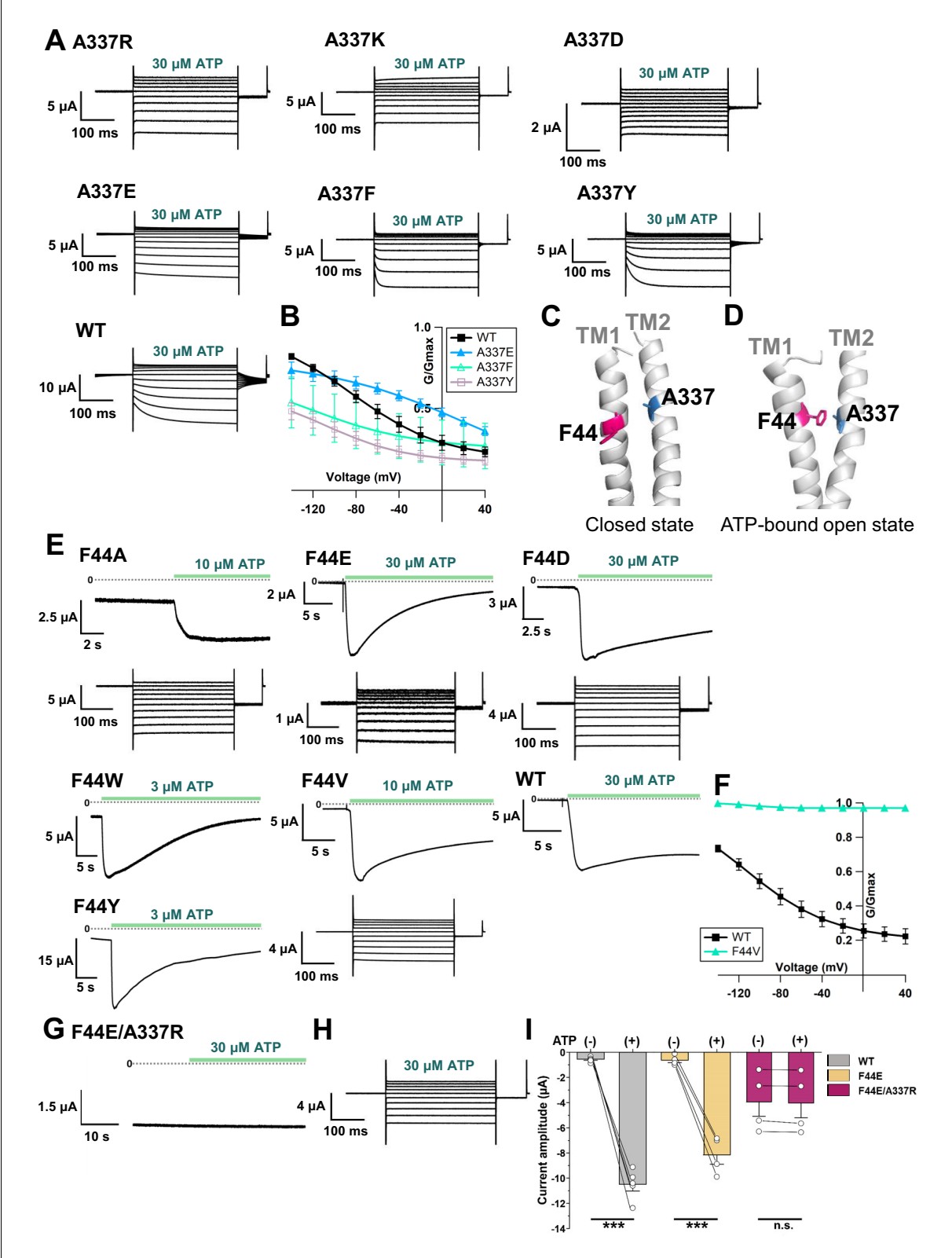

**Figure 7.** Effects of mutations at A337 in TM2 and F44 in TM1 on P2X2 receptor ATP- and voltage-dependent gating. (**A**) Representative current traces of single amino acid mutants at the position of A337 in the presence of 30 μM ATP, in response to voltage step pulses from +40 mV to −140 mV, with a holding potential of −40 mV (A337R, A337K, A337D, A337E, A337F, A337Y, and WT, respectively). (**B**) Comparison of G-V relationships between WT (black filled square), A337E (blue filled triangle), A337F (turquoise open triangle), and A337Y (purple open square) for 30 μM ATP (n = 3), from tail

*Figure 7 continued on next page*

*Figure 7 continued*

current analysis at −60 mV. Normalization was based on the maximum conductance in the highest [ATP] (300 µM) for each construct. (**C, D**) Side view structure of the position of F44 (magenta) and A337 (blue) in the closed (**C**) and ATP-bound open (**D**) state, respectively. (**E**) Representative current traces of single amino acid mutants at the position of F44 upon application of various [ATP] (F44A, F44W, F44Y, F44E, F44D, and WT, respectively; n = 3–6 for each mutant). (**F**) G-V relationship comparison between WT (black filled square) and F44V (turquoise filled triangle) for 10 µM ATP (n = 3), showing that this mutant was equally active at all recorded voltages and was far less sensitive to voltage than wildtype. Normalization was based on the maximum conductance in the highest [ATP] (300 µM) for each construct. (**G, H**) Representative current traces of F44E/A337R upon ATP (**G**) and voltage (**H**) application. (**I**) Comparison of current amplitude of WT, F44E, and F44E/A337R before and after ATP application (***p≤0.001, p=0.00007 for WT, and p=0.00095 for F44E, paired t-test [n = 4–5]). All error bars are ± s.e.m. centered on the mean. Source data are provided in *Figure 7—source data 1*. Statistical analysis data are provided in *Figure 7—source data 2*.

The online version of this article includes the following source data and figure supplement(s) for figure 7:

**Source data 1.** Effects of mutations at A337 in TM2 and F44 in TM1 on P2X2 receptor ATP- and voltage-dependent gating.
**Source data 2.** Statistical analysis to support graph in *Figure 7I*.
**Figure supplement 1.** Effect of swapped mutation F44A/A337F.

Based on the results from VCF recording, mutagenesis experiments, and the homology modeling of *r*P2X2 in the open state upon ATP binding, it was shown that F44 moves into close proximity to the converged electric field at A337 and I341 (*Figure 8A,B*). In the presence of ATP, voltage-dependent conformational changes occur, possibly at or around the position of A337 and F44, giving influence to the interaction between A337 and F44, which is critical for stabilizing the open state. Results of this study show that the origin of the voltage-dependent gating of P2X2 in the presence of ATP is possibly the voltage dependence of the interaction between A337 and F44 within the converged electric field (*Figure 8*).

## Discussion

The present study aims at defining the roles of the TM domains of the P2X2 receptor in complex gating by [ATP] and voltage, using VCF with a genetically incorporated fUAA probe (Anap) and mutagenesis. The following findings were obtained.

### Detection of fast F changes with a linear voltage dependence at A337 and I341

We analyzed 96 mutants by VCF and detected a voltage-dependent $\Delta F_{Fast}/F$ at A337 and I341 in TM2. It was very fast and showed a linear voltage dependence within the voltage range under study. The change could be well interpreted to be due to an electrochromic effect, indicating that there is an electric field convergence at both positions, which are located adjacent to each other.

An electrochromic signal is an intrinsic property exhibited by voltage-sensitive fluorescent dyes or electrochromic probes to directly detect transmembrane potentials (*Loew, 1982*; *Zhang et al., 1998*). By standard use of electrochromic probes in a lipid bilayer, it is difficult to detect the electrical potential that directly acts on the voltage-sensing machinery of membrane proteins (*Asamoah et al., 2003*). This is because the local electric field at a certain position in the lipid bilayer is not steep enough. On the other hand, previous VCF studies on the *Shaker* K$^+$ channel, using modified electrochromic probes (*Asamoah et al., 2003*), and on the M$_2$ muscarinic receptor, using TMRM (*Dekel et al., 2012*), showed that an electrochromic signal could be observed when the fluorophore is directly attached to a specific position within the ion channel/receptor. These studies stated that this phenomenon did not report conformational changes of the protein at the site of fluorophore attachment, but rather implied that there is an electric field convergence if the electrochromic signal is observed only at positions adjacent to each other (*Asamoah et al., 2003*; *Dekel et al., 2012*). This observed electrochromic signal might indicate the possible location of a voltage sensor (*Asamoah et al., 2003*; *Dekel et al., 2012*). Further studies are certainly required to test this possibility.

An almost linear F-V relationship which might originate from the electrochromic signal was also reported from VCF studies in a canonical VSD-containing membrane protein called hTMEM266, labeled with MTS-TAMRA. The observed $\Delta F_{Fast}/F$ was, however, explained rather differently. Even though the $\Delta F_{Fast}/F$ was observed at most of the substituted positions located in the S3-S4 linker

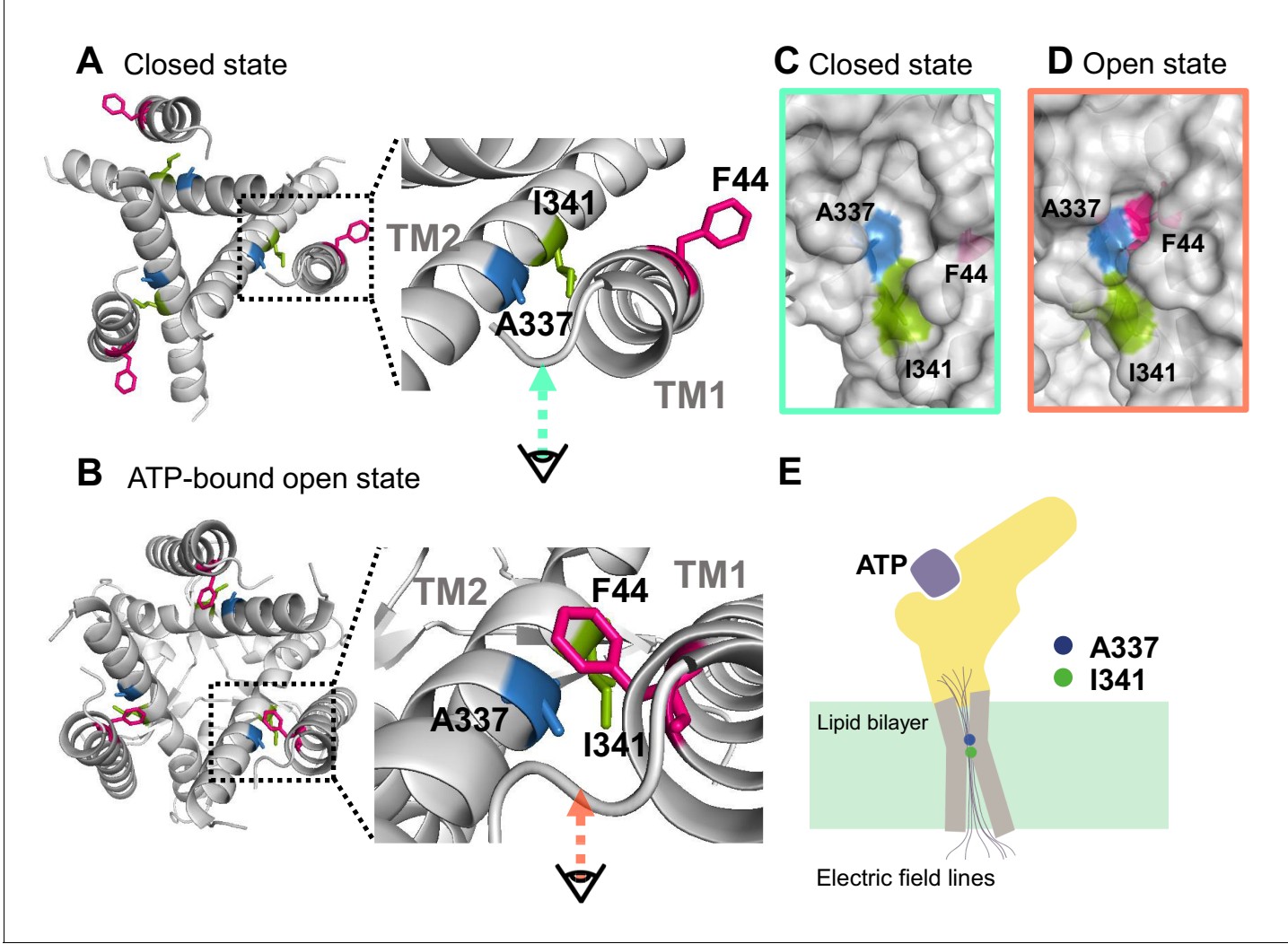

**Figure 8.** Proposed initiation mechanisms of P2X2 receptor complex gating. (**A, B**) Top view structure of the P2X2 receptor in the closed (**A**) and ATP-bound open state (**B**). Depicted are the proposed initiation mechanisms of P2X2 receptor complex gating as follows. (1) The electric convergence at A337 and I341; (2) F44 moves toward A337 in the TM2 domain upon ATP binding; (3) hyperpolarization-induced structural rearrangements around A337 in TM2; (4) interaction between A337 and F44 in the ATP-bound open state is thought to be under the influence of the converged electric field. (**C, D**) Side view of the surface representation of the crevices in the converged electric field in the closed state (**C**) and ATP-bound open state (**D**). (**E**) A schematic illustration of the focused electric field at A337 and I341. The ion permeation pathway is not depicted in this scheme.

and the top of the S4 segment, it was stated that $\Delta F_{Fast}/F$ was not due to a direct electrochromic effect but instead was associated with rapid voltage-dependent conformational changes on a µs time scale (*Papp et al., 2019*). In the case of hTMEM266, it is difficult to surmise that the fast change detected at many positions is due to electrochromic effect, because it suggests an unlikely possibility that the electric field is converged at various positions. Conversely, in the P2X2 receptor, there were only two adjacent positions which exclusively showed clear $\Delta F_{Fast}/F$ and a linear F-V relationship.

In the hTMEM266 study, it was also a concern whether TAMRA-MTS could report an electrochromic signal, because there were not any previous findings to support this case. There were also no reports of electrochromic signals recorded using Anap as fluorophore. Anap has only been reported as an environmental sensitive fluorophore (*Lee et al., 2009*; *Chatterjee et al., 2013*). There are no reports that Anap is an electrochromic fluorophore, unlike the case of the modified fluorophore used in *Shaker* Kv studies, which has been reported to have electrochromic properties (*Zhang et al.,*

*1998*; *Asamoah et al., 2003*). On the other hand, studies on the M$_2$ muscarinic receptor did not discuss TMRM fluorophore properties, but still concluded that the observed fast F change with linear F-V originated from the electrochromic signal (*Dekel et al., 2012*). Even though other possibilities still remain, the most straightforward explanation to interpret the results observed in this study is that the very fast and linearly voltage-dependent fluorescence changes of Anap at A337 and I341 are associated not with the conformational changes of the P2X2 protein but presumably with the electrochromic signal. Consequently, the results show that there is an electric field convergence at these positions, which could give us a clue about the possible location of the voltage sensor in the P2X2 receptor.

We observed that $\Delta F_{Fast}/F$ changed with voltage in both the closed and ATP bound-open states, implying the presence of the focused electric field in both states at the position of A337. $\Delta F_{Fast}/F$ was larger in the absence than in the presence of ATP. One simple interpretation is that the focused electric field is stronger in the absence of ATP. However, as the environment of the fluorophore is different in the absence and presence of ATP, due to the difference in the crevice shape (*Figure 8C,D*), it is not possible to directly compare the strength of the focused electric field in the two states.

Some Cys accessibility studies were performed on the P2X2 receptor in the TM2 domain, to analyze the ATP-evoked gating mechanism (*Li et al., 2008*; *Kracun et al., 2010*; *Li et al., 2010*). A337 Cys mutants were first reported to be not modified by MTSET both in the presence or absence of ATP, indicating that this residue is not involved either in the pore lining region in the open state or in the gate of P2X2, unlike I341 Cys mutants, which were modified by MTSET (*Li et al., 2008*). Meanwhile in another study using Ag$^+$, a smaller thiol-reactive ion with higher accessibility, A337C was modified both in the absence and presence of ATP, as well as I341C (*Li et al., 2010*). These results suggest that a narrow water phase penetrates down to these positions, which is consistent with the results in this study that there is a focused electric field at the position of A337 and I341.

## Detection of slow F change with non-linear voltage dependence at A337 of K308R mutant

We obtained data supporting voltage-dependent conformational rearrangements occurring at or around the position of A337, by analyzing the mixed Anap fluorescence signal changes, which contain both $\Delta F_{Fast}/F$ and $\Delta F_{Slow}/F$, in the presence of an additional mutation of K308R in A337Anap. K308 is located in the ATP binding site and was reported to be important not only for ATP binding but also for the gating of the P2X2 receptor (*Ennion et al., 2000*; *Jiang et al., 2000*; *Roberts et al., 2006*; *Cao et al., 2007*). In VCF analysis, a high expression level is needed to detect $\Delta F$ successfully, because of the background fluorescence. However, high expression makes the P2X2 channel activate even in the absence of ATP and also even at depolarized potentials, that is, the G-V relationship is shifted to a depolarized potential, which makes the voltage-dependent activation upon hyperpolarization unclear. To overcome this problem, we introduced the K308R mutation, which shifts the G-V relationship in the hyperpolarized direction, with much reduced activity at depolarized potentials (*Keceli and Kubo, 2009*). By introducing the K308R mutation, we could observe voltage-dependent gating better and succeeded in recording the slow and voltage-dependent $\Delta F$ at A337 (*Figure 5*).

In addition, the $\Delta F_{Slow}$ component was observed only at hyperpolarized potentials and in the presence of ATP (*Figure 5F-J*; *Figure 6*). Also, the $F_{Slow}$-V and G-V overlapped well, showing that $\Delta F_{Slow}$ reflects the hyperpolarization-induced structural rearrangements around the position of A337 (*Figure 5E*; *Figure 5J*). A337 in TM2 is indeed in the converged electric field, as shown by the linear F-V relationship of the $\Delta F_{Fast}$ component (*Figure 5D*). It is possible that the voltage-dependent $\Delta F_{Slow}$ observed at A337 might not directly reflect the presumably very fast movement of the 'voltage-sensor' in response to the change in membrane voltage but might report the secondary structural rearrangements (*Figure 5A,F*, *Figure 5—figure supplement 2*, *Figure 6A*).

The electric field convergence takes place at residues located in TM2, as shown by the observed Anap fluorescence changes at A337 and I341 (*Figure 3*). Minor changes were also observed at L334 and L338 by follow-up experiments with SIK inhibitor, after the initial screening (*Figure 3—figure supplement 1*). Based on these results, the most frequently observed and the largest electrochromic signal was from A337. Voltage-dependent structural changes were also detected at or around A337. Although a possibility still remains that other residues are also involved but the changes were

undetected due to technical limitations, all the results so far support the interpretation that the main focus for the voltage-sensing mechanism in the P2X2 receptor lies at or around A337.

## Interaction between A337 in TM2 and F44 in TM1 in the converged electric field

The specific function of each transmembrane domain of the P2X receptor had been defined before the crystal structure was solved but the information as to the role of each TM in P2X2 voltage-dependent gating is limited. TM1 is shown to play a role in the binding-gating process, as mutations in this region alter the agonist selectivity and sensitivity of channel gating (*Haines et al., 2001*; *Li et al., 2004*; *Stelmashenko et al., 2014*). In contrast, TM2 plays an essential role in permeation (*Nakazawa et al., 1998*; *Khakh and Egan, 2005*) and gating (*Li et al., 2008*).

Mutations of A337 in the present study suggested that this position is critical for complex gating, as mutation to A337F and A337Y altered the channel gating as well as the activation kinetics upon the application of ATP and voltage (*Figure 7A,B*). The counterpart for A337 is most likely the F44 residue in TM1. Based on the homology modeling of P2X2, in the ATP-bound open state, F44 rotates and moves toward TM2, specifically into the proximity of A337 (*Figure 7C,D*). Mutagenesis at the position of F44 showed the importance of F44 to maintaining the open state in the presence of ATP (*Figure 7E,F*). Taken together, the results raised the possibility that F44 serves as a voltage sensor which has quadrupole moment due to its benzene ring (*Dougherty, 1996*). The aromatic side chain of F44 could have a permanent and non-spherical charge distribution (*Dougherty, 1996*) that is expected to respond to voltage changes in the converged electric field.

The artificial electrostatic bridge in the F44E/A337R mutant (*Figure 7G-I*) induced constitutive activity in the absence of ATP and at all recorded voltages, confirming the importance of the interaction for the maintenance of the activated state. As F44 possibly serves as the voltage sensor, this interaction, which takes place in the converged electric field, may be influenced by the change in membrane voltage. The structural rearrangement at F44 is of very high interest, but F44Anap was not functional, further showing the critical role of F44.

There are several types of voltage-sensing mechanism in membrane proteins (*Bezanilla, 2008*): (1) charged residues, as in the case of canonical voltage-gated ion channels (*Jiang et al., 2003b*; *Swartz, 2008*), (2) side-chains that have an intrinsic dipole moment, such as Tyr, as in the case of the $M_2$ muscarinic receptor (*Ben-Chaim et al., 2006*; *Navarro-Polanco et al., 2011*; *Dekel et al., 2012*; *Barchad-Avitzur et al., 2016*), (3) the $\alpha$-helix, with its intrinsic dipole moment, and (4) cavities within the protein structure, filled with free ions.

Based on our results, the interaction between A337 and F44 in the ATP-bound open state is under the influence of the converged electric field (*Figure 8A-E*), and the results of mutagenesis studies (*Figure 7E,F*) suggest that F44 might serve as a voltage-sensor due to its intrinsic quadrupole. The results also clearly demonstrate that there are voltage-dependent structural rearrangements in the proximity of A337 in TM2.

Our interpretation, based on this study, is that F44 possibly has a two-step voltage-dependent rearrangement, after moving into the proximity of A337 upon ATP application. (1) F44 quickly orients to A337 in the open state, which is stabilized better upon hyperpolarization. (2) The stabilization upon hyperpolarization, as the consequence of F44 voltage sensing, results in secondary structural rearrangements. The slow kinetics of fluorescence signal change at A337 (*Figure 5*, *Figure 5—figure supplement 2*, *Figure 6*) might be the result of the second step of the voltage sensing mechanism by F44. Further analysis of the structural dynamics at the position of F44 will help to elucidate the detailed mechanism of the complex gating of the P2X2 receptor.

## Materials and methods

### Key resources table

| Reagent type (species) or resource | Designation | Source or reference | Identifiers | Additional information |
|---|---|---|---|---|

*Continued*

| Reagent type (species) or resource | Designation | Source or reference | Identifiers | Additional information |
|---|---|---|---|---|
| Gene (*Rattus norvegicus*) | *Rattus norvegicus* P2X2 | *Brake et al., 1994* | | |
| Gene (*Ciona intestinalis*) | *Ciona intestinalis* voltage-sensing phosphatase (*Ci*-VSP) | *Sakata et al., 2016* | | |
| Strain, strain background (*Escherichia coli*) | XL1-Blue | Agilent Technologies | | |
| Strain, strain background (*Escherichia coli*) | TG1 | Clontech | | |
| Recombinant DNA reagent | pAnap (plasmid) | Addgene | Plasmid #48696 | cDNA encoding the tRNA synthetase/Anap-CUA |
| Commercial assay or kit | QuikChange II site-directed mutagenesis | Agilent Technologies | 200524 | |
| Commercial assay or kit | mMESSAGE T7 RNA transcription kit | Thermo Fisher Scientific | AM1344 | |
| Commercial assay or kit | mMESSAGE SP6 RNA transcription kit | Thermo Fisher Scientific | AM1340 | |
| Chemical compound, drug | 0.15% tricaine | Sigma-Aldrich | | |
| Chemical compound, drug | Collagenase type 1 | Sigma-Aldrich | | |
| Chemical compound, drug | ATP disodium salt | Sigma-Aldrich | 34369-07-8 | |
| Chemical compound, drug | Anap sodium salt | FutureChem Chemicals | FC-8101 | |
| Chemical compound, drug | HG 9-91−01/SIK inhibitor | MedChem Express | 1456858-58-4 | |
| Chemical compound, drug | Suramin sodium salt | Sigma-Aldrich | 129-46-4 | |
| Chemical compound, drug | PPADS tetrasodium salt | Sigma-Aldrich | P178 | |
| Software, algorithm | Igor Pro 5.01 | Wavemetrics | RRID:SCR_000325 | |
| Software, algorithm | PyMOL Molecular Graphics System ver. 2.3.0 | Schrodinger LLC | RRID:SCR_000305 | |
| Software, algorithm | OriginPro | OriginLab | RRID:SCR_014212 | |
| Software, algorithm | GraphPad Prism 9 | GraphPad Software, Inc. | RRID:SCR_002798 | |
| Software, algorithm | SWISS-MODEL | *Arnold et al., 2006*; *Biasini et al., 2014* | RRID:SCR_018123 | |
| Software, algorithm | Protter protein visualization | *Omasits et al., 2014* | | https://wlab.ethz.ch/protter/start/ |
| Software, algorithm | BioRender | BioRender.com | RRID:SCR_018361 | *Figure 1A* created with BioRender |

## Ethical approval

All animal experiments were approved by the Animal Care Committee of the National Institutes of Natural Sciences (NINS, Japan) and performed obeying its guidelines.

## Molecular biology

Wild type (WT) *Rattus norvegicus* P2X2 (*r*P2X2) receptor cDNA (*Brake et al., 1994*) was subcloned into the BamH1 site of pGEMHE. TAG or any single amino acid mutation and/or double mutations were introduced using a Quikchange site-directed mutagenesis kit (Agilent Technologies). The introduced mutations were confirmed by DNA sequencing. The mMESSAGE T7 RNA transcription kit (Thermo Fisher Scientific) was used to transcribe WT and mutant *r*P2X2 cRNAs from plasmid cDNA linearized by Nhe1 restriction enzyme (Toyobo). The tRNA-synthetase/Anap-CUA encoding plasmid was obtained from Addgene. The salt form of fUAA Anap was used (Futurechem).

*Ciona intestinalis* voltage-sensing phosphatase (*Ci*-VSP) with a mutation in the gating loop of the phosphatase domain (F401Anap) was used as a positive control (*Sakata et al., 2016*). The mMESSAGE SP6 RNA transcription kit (Thermo Fisher Scientific) was used for cRNA transcription of *Ci*-VSP.

## Preparation of *Xenopus laevis* oocytes

As an anesthetic agent, 0.15% tricaine (Sigma-Aldrich) was used for *Xenopus laevis* before surgical operation for isolation of oocytes. After the final collection, the frogs were humanely sacrificed by decapitation. Follicular membranes were removed from isolated oocytes by collagenase treatment (2 mg ml$^{-1}$; type 1; Sigma-Aldrich) for 6.5 hr. Oocytes were then rinsed and stored in frog Ringer's solution (88 mM NaCl, 1 mM KCl, 2.4 mM NaHCO$_3$, 0.3 mM Ca(NO$_3$)$_2$, 0.41 mM CaCl$_2$, 0.82 mM Mg$_2$SO$_4$, and 15 mM HEPES pH 7.6 with NaOH) containing 0.1% penicillin-streptomycin at 17°C.

## Channel expression and electrophysiological recording of *r*P2X2

*Xenopus* oocytes injected with 0.5 ng of WT *r*P2X2 cRNA and incubated for 2 days at 17°C showed a high expression level phenotype of WT *r*P2X2 that has less voltage dependence than those of low expression level of P2X2 (I < 4.0 µA at −60 mV) (*Fujiwara and Kubo, 2004*). To achieve low expression level, oocytes were injected with 0.05 ng of WT *r*P2X2 cRNA and incubated for 1–2 days. For *r*P2X2 mutants, oocytes were injected with 0.5 ng–2.5 ng of cRNA and incubated for 1–3 days, depending on the desired expression level.

Voltage clamp for macroscopic current recording was performed with an amplifier (OC-725C; Warner Instruments), a digital-analogue analogue-digital converter (Digidata 1440, Molecular Devices), and pClamp10.3 software (Molecular Devices). In TEVC recording, borosilicate glass capillaries (World Precision Instruments) were used with a resistance of 0.2–0.5 MΩ when filled with 3 M KOAc and 10 mM KCl. P2X2 bath solution contained 95.6 mM NaCl, 1 mM MgCl$_2$, 5 mM HEPES, and 2.4 mM NaOH at pH 7.35–7.45. Ca$^{2+}$ was not included in the bath solution in order to avoid the inactivation of the receptor and secondary intracellular effects, for example, activation of Ca$^{2+}$ dependent chloride channel currents, (*Ding and Sachs, 2000*).

ATP disodium salt (Sigma-Aldrich) was prepared in various concentrations (1 µM, 3 µM, 10 µM, 30 µM, 100 µM, 300 µM, 1 mM, and 3 mM) by dissolving it in the bath solution. For recording using step-pulse protocols, ATP was applied in two ways, depending on the purpose of the experiments and the phenotype of the mutants. (1) Direct application using a motorized pipette (Gilson pipetman), which was set to exchange the whole bath solution with a ligand-based solution. 2000 µL (five times larger than the bath volume) of ligand-based solution was applied. (2) Perfusion of a recording chamber using a perfusion system set (ISMATEC pump). In both cases, overflowed bath solution was continuously removed using a suction pipette by negative air pressure. Oocytes were held at −40 mV and voltage step pulses were applied in the range from +40 mV to −140 mV. For TEVC recordings with P2X2 non-specific blockers, oocytes were held at +20 mV, and voltage step pulses were applied in the range from +40 mV to −160 mV. 300 µM of Suramin sodium salt and 300 µM or 1 mM of PPADS tetrasodium salt (Sigma-Aldrich) was directly applied by motorized pipette. First, 10 or 300 µM ATP was applied to elicit the ATP-evoked current, and subsequently in the same cell, either suramin or PPADS was applied. Tail currents were recorded at −60 mV to measure conductance-voltage (G-V) relationships. Recordings were performed at room temperature (24 ± 2°C).

## Expression of Anap-incorporated *r*P2X2 and *Ci*-VSP

For functional expression of channels with incorporated Anap, 1.25 ng of cDNA encoding the tRNA synthetase/Anap-CUA pair was injected into the nucleus of defolliculated *Xenopus* oocytes located

in the center of the animal pole (*Kalstrup and Blunck, 2013*). Oocytes were then incubated for 24 hr at 17°C to allow tRNA transcription and synthetase expression. The subsequent step was performed with minimization of light exposure, which otherwise may have excited the fluorophore. Either 1.4–12.6 ng of *r*P2X2 cRNA or 8.2 ng of *Ci*-VSP cRNA in which the target site was mutated to a TAG codon, was co-injected with 23 nL of 1 mM Anap. Oocytes were incubated in frog Ringer's solution (containing 0.1% penicillin-streptomycin) for 1–3 days (*r*P2X2) or 3–5 days (*Ci*-VSP) depending on the desired expression level. In the absence of either tRNA synthetase/Anap-CUA plasmid or fUAA Anap, no channel expression was detected in *r*P2X2 Anap mutants, confirming that functional channels were expressed only when they had successfully incorporated fUAA.

## SIK inhibitor application

HG 9-91−01/SIK inhibitor (MedChem Express) was dissolved in DMSO to make a stock solution of 10 mM and kept as aliquots at −80°C. SIK inhibitor was diluted before use with RNase-free water (Otsuka) into certain concentrations for injection to oocytes. Various concentrations of SIK inhibitor were injected into oocyte nuclei to determine the most effective concentration to improve the optical recording of VCF-fUAA. SIK inhibitor was mixed and co-injected with either (1) tRNA synthetase/Anap-CUA plasmid (nuclear injection) or (2) cRNA+Anap (cytoplasmic injection). 300 nM was defined as the amount of the co-injected SIK inhibitor in the mixed solution. For instance, the actual concentration of SIK inhibitor is 600 nM for 1:1 mixture with 2.5 ng tRNA synthetase/Anap-CUA plasmid. As the volume of the oocyte nucleus is ~40 nL, and it can tolerate 15–20 nL of injected volume (*Lin-Moshier and Marchant, 2018*), the final concentration of SIK inhibitor inside the oocyte nucleus was ~150 nM.

First of all, *Ci*-VSP F401Anap was used to confirm reproducible effects in the initial optimization experiments. The most effective concentration of SIK inhibitor was determined to be 300 nM. Next, 300 nM of SIK inhibitor was co-injected into either the nucleus or cytoplasm of the oocytes, which were then incubated for different periods of time. This resulted in three test groups: (1) nuclear injection with 2 days incubation, (2) nuclear injection with 3 days incubation, and (3) cytoplasmic injection with 2 days incubation. Cytoplasmic injection needs concentration adjustment, since the volume of an oocyte is ~1 μL. To make the concentration inside the oocyte 150 nM, the injected concentration was 3 μM. Control groups consisted of non-treated oocytes, incubated for either 2 or 3 days.

A follow-up confirmation experiment was done using the P2X2 A337Anap/R313W mutant, after the optimum concentration, injection method, and incubation period were determined from the *Ci*-VSP experiment. 300 nM of SIK inhibitor was co-injected into the nucleus of the oocyte. Oocytes were then incubated for 2–3 days, after subsequent cytoplasmic co-injection of channel cRNA and Anap.

## Voltage-clamp fluorometry (VCF) recording

Oocytes for VCF-fUAA recording needed to be shielded from light exposure. Oocytes were placed in a recording chamber with the animal pole facing upward. For ATP-evoked current recording, a gap-free protocol was applied, with the holding potential at −80 mV. ATP was applied by perfusion system as described above. For voltage-evoked current recording, oocytes were held at +20 mV or at −40 mV in some cases. The step pulses were applied from +40 mV to −140 mV, +40 mV to −160 mV, or +80 mV to − 160 mV.

Two recordings (ATP application and voltage application) were performed separately in different oocytes. VCF recordings in the absence and presence of ATP using voltage step pulses, for some mutants (A337Anap, R313F/A337Anap, R313W/A337Anap, and K308R/A337Anap), were performed in the same oocytes.

For voltage step application, ATP was applied directly. As bath volume was measured to be 600 μL, 20 μL ATP of 30 times higher concentration was applied directly to the bath solution. For *Ci*-VSP voltage-clamp recording, cells were clamped at −60 mV and the step pulses were applied from −80 mV to +160 mV every 3 s.

Fluorometric recordings were performed with an upright fluorescence microscope (Olympus BX51WI) equipped with a water immersion objective lens (Olympus XLUMPLAN FL 20x/1.00) to collect the emission light from the voltage-clamped oocytes. The light from a xenon arc lamp (L2194-01, Hamamatsu Photonics) was applied through a band-pass excitation filter (330–360 nm for Anap).

In the case of the excitation of Anap to minimize photobleaching during ATP-application recording, the intensity of the excitation light was decreased to 1.5% with ND filters (U-25ND6 and U-25ND25 Olympus), whereas, for step-pulse recording, the intensity of the excitation light was decreased to 6% (U-25ND6 Olympus). Emitted light was passed through band pass emission filters (Brightline, Semrock) of 420–460 nm and 460–510 nm (*Lee et al., 2009*; *Sakata et al., 2016*). The emission signals were detected by two photomultipliers (H10722-110; Hamamatsu Photonics). The detected emission intensities were acquired by a Digidata 1332 (Axon Instruments) and Clampex 10.3 software (Molecular Devices) at 10 kHz for ATP application and 20 kHz for voltage application. In the case of *Ci*-VSP, the detected emission was acquired at 10 kHz. To improve the signal-to-noise ratio, VCF recording during step-pulse protocols was repeated 20 times for each sample for P2X2 in the presence of ATP, five times in the absence of ATP, and three times for *Ci*-VSP. Averaged data were used for data presentation and analysis.

## Data analysis

Two electrode voltage-clamp data were analyzed using Clampfit 10.5 software (Molecular Devices) and Igor Pro 5.01 (Wavemetrics). Analyses of conductance-voltage (G-V) relationship of P2X2 were obtained from tail current recordings at −60 mV and fitted to a two-state Boltzmann equation using Clampfit:

$$I = I_{min} + \frac{I_{max} - I_{min}}{1 + e^{\frac{ZF}{RT}\left(V - V_{\frac{1}{2}}\right)}} \tag{1}$$

where $I_{min}$ and $I_{max}$ are defined as the limits of the amplitudes in fittings, $Z$ is defined as the effective charge, $V_{1/2}$ is the voltage of half activation, $F$ is Faraday's constant, and $T$ is temperature in Kelvin.

In the case of P2X2, normalized conductance-voltage (G-V) relationships were plotted using:

$$G/G_{max} = I/I_{min} = 1 - \left(1 + e^{ZF\left(V - V_{1/2}\right)/RT}\right)^{-1}\left(1 - I_{max}/I_{min}\right) \tag{2}$$

In the case of VCF data, the gradual decline of fluorescence recording traces due to photobleaching was compensated by subtracting the expected time-lapse decrease calculated from the trace's bleaching rate (R) by assuming that the fluorescence is linear. Arithmetic operations were performed by Igor Pro 5.01 for ATP-evoked fluorescent signals.

$$\text{Compensated data} = \text{Recorded F data} + R^* \text{ point number} \tag{3}$$

In the case of fluorescence traces from voltage application for both P2X2 and *Ci*-VSP, arithmetic operations were performed by Clampfit.

$$[\text{Compensated trace}] = [\text{Recorded F trace}] \times (1 - (R \times [\text{time}])) \tag{4}$$

where [time] is the value of the point given by Clampfit. All the compensated traces were then normalized by setting each baseline level (F signal at −40 mV or at +20 mV depending on the holding potential) to be 1, to calculate the % F change (ΔF/F; ΔF = $F_{-160mV}$ − $F_{baseline}$; F = $F_{baseline}$). The fraction of $\Delta F_{Slow}/F$ was calculated from the equation:

$$\Delta F_{steady-state}/F = \Delta F_{fast}/F + \Delta F_{slow}/F; F = F_{baseline} \tag{5}$$

The data were expressed as mean ± s.e.m. with n indicating the number of samples.

## Statistical analysis

Statistical analysis was performed by either one-way ANOVA, two-sample t-test, or paired t-test. Following one-way ANOVA, Tukey's post-hoc test was applied. The data were expressed as mean± s. e.m. with n indicating the number of samples. Values of $p < 0.05$ were defined as statistically significant. *, **, ***, **** denote values of $p < 0.05, 0.01, 0.001, 0.0001$, respectively. All the statistical analysis and the bar graphs were performed and generated either with OriginPro (OriginLab) or GraphPad Prism 9.

### Three-dimensional structural modeling of rat P2X2

Homology modeling was performed using a web-based environment for protein structure homology modeling SWISS-MODEL (*Arnold et al., 2006*; *Biasini et al., 2014*), based upon the amino acid sequence of *r*P2X2 (NM_053656) and the crystal structure of *h*P2X3 (Protein Data Bank accession number 5SVJ and 5SVK for closed and ATP-bound open states, respectively) (*Mansoor et al., 2016*). All the structural data presented in this study were generated using PyMOL molecular graphics system ver. 2.3.0 (Schrodinger LLC). Protein visualization was generated using Protter (*Omasits et al., 2014*).

## Acknowledgements

The authors thank Dr. Sakata S and Prof. Okamura Y (Osaka University, Graduate School of Medicine) for the guidance of VCF experiments, Dr. Shimomura T and all members in Kubo Laboratory for discussion, Ms. Naito C for technical support, and Dr. Collins A (Saba University, School of Medicine, Dutch Caribbean) for editing the manuscript.

## Additional information

### Funding

| Funder | Grant reference number | Author |
| --- | --- | --- |
| Japan Society for the Promotion of Science | KAKENHI 17H04021 | Yoshihiro Kubo |
| Japan Society for the Promotion of Science | KAKENHI 20H03424 | Yoshihiro Kubo |
| Daiko Foundation | | Rizki Tsari Andriani |

The funders had no role in study design, data collection and interpretation, or the decision to submit the work for publication.

### Author contributions

Rizki Tsari Andriani, Conceptualization, Data curation, Formal analysis, Writing - original draft, Writing - review and editing; Yoshihiro Kubo, Conceptualization, Supervision, Funding acquisition, Project administration, Writing - review and editing

### Author ORCIDs

Rizki Tsari Andriani https://orcid.org/0000-0002-9242-469X
Yoshihiro Kubo https://orcid.org/0000-0001-6707-0837

### Ethics

Animal experimentation: All animal experiments were approved by the Animal Care Committee of the National Institutes of Natural Sciences (NINS, Japan) and performed obeying its guidelines.

### Decision letter and Author response

Decision letter https://doi.org/10.7554/eLife.65822.sa1
Author response https://doi.org/10.7554/eLife.65822.sa2

## Additional files

### Supplementary files

• Supplementary file 1. List of introduced TAG mutations in P2X2 receptor for VCF analysis. Mutations were introduced one at a time into 96 positions within the extracellular domain (ECD) near the ATP-binding site and extracellular linker, transmembrane domains (TMs), intracellular N-terminal, and intracellular C-terminal. ATP application ranging from 10 μM, 30 μM, or 100 μM unless

otherwise stated. (+) indicates there was either ATP-evoked fluorescence (F) signal change, voltage-evoked F change, ATP-evoked current (I) change, or voltage-evoked I change. (-) indicates negative results. (**) indicates mutants which have a very low expression level, so that the reliable VCF analysis could not be undertaken. (***) indicates fast current decay. (–) indicates that the subsequent recording could not be performed, as a result of fast current decay. (n.d.) indicates not determined.

- Transparent reporting form

## Data availability

All data generated or analysed during this study are included in the manuscript and supporting files. Source data files have been provided for Figure 1, Figure 1—figure supplement 1, Figure 2, Figure 2—figure supplement 1, Figure 3, Figure 3—figure supplement 1, Figure 4, Figure 4—figure supplement 1, Figure 5, Figure 5—figure supplement 1, Figure 5—figure supplement 2, Figure 6, Figure 6—figure supplement 1, and Figure 7.

The following previously published datasets were used:

| Author(s) | Year | Dataset title | Dataset URL | Database and Identifier |
|---|---|---|---|---|
| Mansoor SE, Lu W, Oosterheert W, Shekhar M, Tajkhorshid E, Gouaux E | 2016 | 5SVJ Crystal structure of the ATP-gated human P2X3 ion channel in the closed, apo state | https://www.rcsb.org/structure/5svj | RCSB Protein Data Bank, 5SVJ |
| Mansoor SE, Lu W, Oosterheert W, Shekhar M, Tajkhorshid E, Gouaux E | 2016 | 5SVK Crystal structure of the ATP-gated human P2X3 ion channel in the ATP-bound, open state | https://www.rcsb.org/structure/5svk | RCSB Protein Data Bank, 5SVK |
| Brake AJ, Wagenbach MJ, Julius D | 1994 | NM_053656 Amino acid sequence of rat P2X2 | https://www.ncbi.nlm.nih.gov/nuccore/NM_053656 | NCBI GenBank, NM_053656.3 |

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
