## [Decision Letter]

**Acceptance summary:**

This study will be of broad interest to ion channel researchers interested in understanding the fundamental mechanisms of voltage-sensing. The researchers use a novel approach to determine the mechanism of voltage-sensing in a channel that lacks a canonical voltage-sensing domain.

**Decision letter after peer review:**

Thank you for submitting your article "Voltage-clamp fluorometry analysis of structural rearrangements of ATP-gated channel P2X2 upon hyperpolarization" for consideration by *eLife*. Your article has been reviewed by 3 peer reviewers, including Baron Chanda as the Reviewing Editor and Reviewer #1, and the evaluation has been overseen by Richard Aldrich as the Senior Editor. The following individual involved in review of your submission has agreed to reveal their identity: Mufeng Li (Reviewer #3).

Essential revisions:

1. In many instances, it is not clear whether the current is specific or non-specific oocyte currents. It is important that the authors use a specific blocker of P2X2 receptor to demonstrate that these are channel specific currents.

2. The current and fluorescence of all the positions listed in table 1 should be shown. At the very least, the authors should show for the positions where they were able to observe voltage-dependent fluorescence response.

3. The interaction between A337 and Phe 44 as a primary determinant of voltage-sensing is a bit confounding. None of them are charged and it is not clear what makes this interaction voltage-sensitive.

4. For electrochromic response, the strength of the local electric field can be estimated using the slope of linear Fluorescence voltage relationship. Is this slope dependent on the expression level?

5. Two related issues raised by two reviewers which should be addressed textually.

The non-canonical voltage sensor in P2X2 channels is an intriguing question. This study suggests a voltage dependent conformational rearrangement in TM2, which is a significant finding. Nevertheless, the voltage sensor for this to happen is not identified. F44 could serve as the voltage sensor with its aromatic dipole moment. However, if F44 is the voltage sensor the fluorescence changes at A337 should measure the movement of F44 in response of voltage directly. How fast should this movement be? Isn't the time course of the fluorescence decrease in Figure 5 and 6 too slow for this process? If the fluorescence decrease, in Figure 5 and 6, is due to a slow overall structural rearrangement as the consequence of F44 sensing voltage, what is the expected voltage dependence of the changes in dipole moment?

I am not convinced that "the main focus for the voltage-sensing mechanism in the P2X2 receptor lies at or around A337." (line 491-492) As the authors mentioned, the florescence changes at A337 and I341 are small and only occur in some oocytes tested, so it is possible that the changes at other residues went undetected. There is no flaw in the experimental design, but there are limitations imposed by the method and the behavior of the protein. P2X receptors are trimers, which means they have less fluorophores per channel compared to the tetramers or pentamers. In order to obtain decent fluorescence signal, channels were expressed at very high density in those experiments. Unfortunately, high channel density is known to diminish the voltage dependence of P2X receptor channels, making it difficult to detect voltage-dependent changes.

6. Another related issue raised by both reviewers.

Fluorescence changes at A337 were reduced by ATP (Figure 4). Since a higher ATP concentration is correlated with a larger current (Figure 4), fluorescence changes might be reduced by increases of currents. In Figure 5 and 6 the time dependent and non-linear fluorescence changes were accompanied by a time dependent current increase. In these experiments was the time dependent fluorescence decrease induced by the time dependent current increase, or the current increase was a result of the conformational change that was measured by the time dependent fluorescence decrease?

After a careful examination of the represented traces shown in figure 4 and figure 6, I am not sure one should directly compare the ΔF/F in the absence and presence of ATP to make the conclusion that "the focused electric field at the position of A337 is stronger in the absence of ATP than in the presence of ATP." (Line 491-492) The noise of the fluorescence signal is always noticeably smaller in the presence of ATP suggesting a possible change in the F. In the hP2X3 receptor closed state structure, residue equivalent to A337 is more than 5Å from that equivalent to F44 in TM1. This part of TM2 is separated from TM1 and Anap incorporated at A337might be surrounded by some lipids in the closed state. In the presence of ATP, when channel opens, this part of TM2 gets close to TM1, where A337 and F44 can interact. Anap incorporated at A337 could be shielded by TM1 in the open state. As one can imagine, those structural rearrangements during channel opening could very likely change the environment for Anap at A337. It would not be meaningful to directly compare ΔF/F recorded in different environment where the fluorophores have inconsistent behavior. I do understand VCF recording was repeated 20 times for each sample in the presence of ATP and only 5 times in the absence of ATP. The difference of noise could also result from the average of different number of traces. Either way, I would suggest the authors to address the difference.

7. The small size of the slow ΔF shown in figure 5 could potentially lead to inaccurate data analysis. It would be better if there are more evidence to support that the slow ΔF results from voltage dependent conformational change in the protein. As previously reported by the Kubo lab, the voltage dependent activation time constant varies with different ATP concentrations. It will be a great supporting evidence if the time constant of the slow fluorescence change vary with ATP concentration.

8. Why was I341 not studied further? If it is because of technical difficulties with small signals, then we cannot rule out its role in voltage sensing and it should be discussed. Additionally, previous findings from Kubo's lab have demonstrated that mutations at T339 and G342 can completely abolish the voltage dependence, suggesting possible involvement of those residues as well.

9. Please provide numbers for the readers to understand how many oocytes were tested and how many showed the change for the statement: "the incidence of fluorescence change detection is also low" (Line 120). The numbers should also be provided later when the authors talk about increase in the incidence of detectable ΔF/F after SIK inhibitor treatment.

10. Is there any correlation between the current size and the detectable ΔF/F? Is it because the channel density is too high for some oocytes to show any voltage dependent change?

11. There is no information about the dose-response of ATP activation for any of the construct tested. What is the rationale for choosing the concentrations used in the experiments?

12. It's shown on Supplementary Table I that no voltage-dependent F change was detected for any residue in TM1 but "n.d." was used to indicate the ATP evoked F change for six residues. What does "n.d." mean exactly and why couldn't it be determined?

---

## [Author Response]

Essential revisions:1. In many instances, it is not clear whether the current is specific or non-specific oocyte currents. It is important that the authors use a specific blocker of P2X2 receptor to demonstrate that these are channel specific currents.

Thank you for alerting us the need for validating this point. We agree with the reviewers that it is important to demonstrate that the currents evoked by ATP in this study are P2X2 channel specific currents. We have added data showing that ATP-evoked currents of the three most important constructs in this study, namely A337Anap, I341Anap and K308R/A337Anap, could be blocked by non-specific blockers of P2X2 receptor, Suramin and PPADS (Figure 1—figure supplement 1; Figure 5—figure supplement 1). The data confirmed that the ATP-evoked currents of these constructs were indeed P2X2 channel specific currents. We could not use a specific blocker of P2X2 (e.g., NF770 or PSB-10211), since they are not commercially available and it takes time to synthesize the specific blocker on demand.

We added sentences to the result section in the revised manuscript as follows:

“ATP-evoked currents of both constructs were inhibited by the P2X2 receptor non-specific blockers, Suramin and PPADS (Figure 1—figure supplement 1), confirming that the currents are indeed P2X2 receptor currents.”

“ATP-evoked current of K308/A337Anap was inhibited by both Suramin and PPADS, the P2X2 non-specific blockers, confirming that the current is indeed a P2X2 receptor current (Figure 5—figure supplement 1).”

2. The current and fluorescence of all the positions listed in table 1 should be shown. At the very least, the authors should show for the positions where they were able to observe voltage-dependent fluorescence response.

Thank you for this suggestion. The current and the fluorescence traces of the positions where we were able to observe voltage-dependent fluorescence response are previously shown in Figure 1 E, F in the manuscript (for A337Anap and I341Anap, respectively). We agree with the reviewers that it is important to show the readers recordings of current and fluorescence of Anap mutant scanning experiments. In the revised version of the manuscript, we added current and fluorescence recordings of some important positions listed in table 1. We covered all the transmembrane regions, as these regions are thought to be potentially important for the non-canonical voltage-sensing mechanism. The new figures are added as Figure 1—figure supplement 2 (for transmembrane 1 region) and Figure 1—figure supplement 3 (for transmembrane 2 region).

We added a sentence to the result section in the revised manuscript as follows:

“Voltage-dependent F changes could not be detected at other scanned positions in TM1, TM2 or other regions (Figure 1—figure supplement 2, 3; Supplementary file 1).”

3. The interaction between A337 and Phe 44 as a primary determinant of voltage-sensing is a bit confounding. None of them are charged and it is not clear what makes this interaction voltage-sensitive.

Thank you for this valuable comments. We agree that the way we explained about the interaction between A337 and F44 was rather vague. From the results of the artificial bridge formation experiments, we showed that the interaction between the A337 position and the F44 position stabilizes the open state of P2X2 receptor (Figure 7G-I). The way of writing that the interaction between A337 and F44 is the primary determinant for voltage-sensing was too strong. Thus, we revised the discussion to re-phrase our interpretation.

First, we previously assumed that this interaction could be voltage sensitive because F44 interacts with A337 in the focused electric field (Figure 3; Figure 7C, D; Figure 7G-I; Figure 8A, B). Second, the interpretation for the interaction between A337 and F44 is that the interaction might be a CH-π interaction which had been found to play an important role in the protein structure stability (Brandl et al., 2001). This correlates well with our experimental data showing that the interaction of A337 and F44 is important for stabilizing the open state of P2X2 receptor.

Voltage sensitivity itself is thought to come from F44 as phenylalanine has quadrupole moment from its aromatic (benzene) ring which resulted in a permanent and non-spherical charge distribution (Dougherty, 1996). Thus, we interpreted the results as follows. ATP binding makes F44 rotates and interacts with A337 in the focused electric field. And, as F44 possibility serves as the voltage sensor, this interaction may be influenced by the change in the membrane voltage.

We revised the Discussion section as follows:

“Taken together, the results raised the possibility that F44 serves as a voltage sensor which has quadrupole moment due to its benzene ring (Dougherty, 1996). The aromatic side chain of F44 could have a permanent and non-spherical charge distribution (Dougherty, 1996) that is expected to respond to voltage changes in the converged electric field.

The artificial electrostatic bridge in the F44E/A337R mutant (Figure 7G – I) induced constitutive activity in the absence of ATP and at all recorded voltages, confirming the importance of the interaction for the maintenance of the activated state. As F44 possibility serves as the voltage sensor, this interaction, which takes place in the converged electric field, may be influenced by the change in membrane voltage. The structural rearrangement at F44 is of very high interest, but F44Anap was not functional, further showing the critical role of F44.”

4. For electrochromic response, the strength of the local electric field can be estimated using the slope of linear Fluorescence voltage relationship. Is this slope dependent on the expression level?

It is true that the strength of the local electric field can be estimated using the slope of linear F-V relationship. However, to convert the electrochromic slope into an electric field strength, a calibration to determine the relative sensitivity between two dyes is needed in in-vitro reconstitution system (e.g., the first one is Anap which is incorporated at a specific site in the protein of interest, and the second one is electrochromic fluorescence dye located freely in the cell membrane). Therefore, it was not possible to solely use the slope of linear F-V relationship without the calibration which gives a correction factor. There is also another factor to be considered. It is the background fluorescence of the oocytes which differs from batch to batch. In the high protein expression case, the slope of VCF recording could look steeper, but this does not necessarily imply that the local electric field is stronger. The steeper slope could be due to a relatively lower level of background fluorescence. Even if the expression level is similarly high, if the background fluorescence is different, the observed slope of linear F-V relationship could appear to be different.

5. Two related issues raised by two reviewers which should be addressed textually.The non-canonical voltage sensor in P2X2 channels is an intriguing question. This study suggests a voltage dependent conformational rearrangement in TM2, which is a significant finding. Nevertheless, the voltage sensor for this to happen is not identified. F44 could serve as the voltage sensor with its aromatic dipole moment. However, if F44 is the voltage sensor the fluorescence changes at A337 should measure the movement of F44 in response of voltage directly. How fast should this movement be? Isn't the time course of the fluorescence decrease in Figure 5 and 6 too slow for this process? If the fluorescence decrease, in Figure 5 and 6, is due to a slow overall structural rearrangement as the consequence of F44 sensing voltage, what is the expected voltage dependence of the changes in dipole moment?

We thank the reviewers for pointing out this vital point. Based on our main findings, there are voltage-dependent structural rearrangements in the proximity of A337 in TM2 (Figure 5). We understand that F44 could serve as the voltage sensor. As F44 rotates into the vicinity of A337 in the presence of ATP, this raised a possibility that F44 serves as voltage sensor by its quadrupole moment due to its benzene ring (Dougherty, 1996). Consequently, this makes the aromatic side chain of F44 to have a permanent and non-spherical charge distribution that is expected to respond to voltage changes in the converged electric field (Dougherty, 1996).

However, as the reviewers pointed out, we also noted that the voltage-dependent fluorescence changes observed at A337 might not necessarily report the presumably fast movement of F44 in response to the change in membrane voltage. As shown in Figure 5—figure supplement 2F, the activation speed of fluorescence in 300 µM was, similar to, not faster than that of the current. Please note that due to the large noise and small fluorescence signal, the trustable fitting for fluorescence traces could only be achieved in -160 mV.

Our interpretation is that F44 possibly has two-step voltage-dependent rearrangements after it underwent a conformational change to the proximity of A337 upon ATP application: (1) F44 quickly orients to A337 in the open state which is stabilized better upon hyperpolarization (2) The stabilization upon hyperpolarization as the consequence of F44 voltage sensing results in secondary structural rearrangements. The relatively slow kinetics of fluorescence signal change at A337 (Figure 5 and 6 in the manuscript) might be the result of the second step of voltage sensing mechanism by F44.

We therefore revised the discussion as follows.

“In addition, the ∆F_Slow_ component was observed only at hyperpolarized potentials and in the presence of ATP (Figure 5F – J; Figure 6). Also, the F_Slow_ – V and G-V overlapped well, showing that ∆F_Slow_ reflects the hyperpolarization-induced structural rearrangements around the position of A337 (Figure 5E; Figure 5J). A337 in TM2 is indeed in the converged electric field, as shown by the linear F – V relationship of the ∆F_Fast_ component (Figure 5D). It is possible that the voltage-dependent ∆F_Slow_ observed at A337 might not directly reflect the presumably very fast movement of the “voltage-sensor” in response to the change in membrane voltage but might report the secondary structural rearrangements (Figure 5A, F, Figure 5—figure supplement 2, Figure 6A).”

“Based on our results, the interaction between A337 and F44 in the ATP-bound open state is under the influence of the converged electric field (Figure 8A – E), and the results of mutagenesis studies (Figure 7E, F) suggest that F44 might serve as a voltage-sensor due to its intrinsic quadrupole. The results also clearly demonstrate that there are voltage-dependent structural rearrangements in the proximity of A337 in TM2.

Our interpretation, based on this study, is that F44 possibly has a two-step voltage-dependent rearrangement, after moving into the proximity of A337 upon ATP application. (1) F44 quickly orients to A337 in the open state, which is stabilized better upon hyperpolarization (2) The stabilization upon hyperpolarization, as the consequence of F44 voltage sensing, results in secondary structural rearrangements. The slow kinetics of fluorescence signal change at A337 (Figure 5, Figure 5—figure supplement 2, Figure 6) might be the result of the second step of the voltage sensing mechanism by F44. Further analysis of the structural dynamics at the position of F44 will help to elucidate the detailed mechanism of the complex gating of the P2X2 receptor.”

I am not convinced that "the main focus for the voltage-sensing mechanism in the P2X2 receptor lies at or around A337." (line 491-492) As the authors mentioned, the florescence changes at A337 and I341 are small and only occur in some oocytes tested, so it is possible that the changes at other residues went undetected. There is no flaw in the experimental design, but there are limitations imposed by the method and the behavior of the protein. P2X receptors are trimers, which means they have less fluorophores per channel compared to the tetramers or pentamers. In order to obtain decent fluorescence signal, channels were expressed at very high density in those experiments. Unfortunately, high channel density is known to diminish the voltage dependence of P2X receptor channels, making it difficult to detect voltage-dependent changes.

We would like to thank the reviewers for the critical comments. We realized that simply stating “the main focus for the voltage-sensing mechanism in the P2X2 receptor lies at or around A337.” in line 491-492 was too strong. It is necessary to mention the possibility that there might be other contributing residues whose fluorescent changes were undetected due to the limitation of the experimental technique we performed. i.e., we agree that there might be undetected electrochromic signals or slow component of fluorescence changes at other residues, and thus we revised the discussion as follows:

“The electric field convergence takes place at residues located in TM2, as shown by the observed Anap fluorescence changes at A337 and I341 (Figure 3). Minor changes were also observed at L334 and L338 by follow up experiments with SIK inhibitor, after the initial screening (Figure 3—figure supplement 1). Based on these results, the most frequently observed and the largest electrochromic signal was from A337. Voltage-dependent structural changes were also detected at or around A337. Although a possibility still remains that other residues are also involved but the changes were undetected due to technical limitations, all the results so far support the interpretation that the main focus for the voltage-sensing mechanism in the P2X2 receptor lies at or around A337.”

6. Another related issue raised by both reviewers.Fluorescence changes at A337 were reduced by ATP (Figure 4). Since a higher ATP concentration is correlated with a larger current (Figure 4), fluorescence changes might be reduced by increases of currents. In Figure 5 and 6 the time dependent and non-linear fluorescence changes were accompanied by a time dependent current increase. In these experiments was the time dependent fluorescence decrease induced by the time dependent current increase, or the current increase was a result of the conformational change that was measured by the time dependent fluorescence decrease?

Thank you for pointing out another vital point in this study. To address this issue, we performed additional experiments by using NMDG-based solution in which current amplitude is reduced. By applying 300 µM ATP to an NMDG-based solution, we performed VCF experiment of K308R/A337Anap construct. After the application of 300 µM ATP, we observed Anap fluorescence signal changes when there was very small current, as shown in Author response image 1. The results suggests that fluorescence (F) change was not the result of the current. However, in this set of experiment we could not observe a clear non-linear F change. This could probably due to the variation from batch to batch of the oocyte, i.e. if the background fluorescence is high it is hard to observe the non-linear F change.

Next, we performed a series of experiments in the same cell by using different bath solutions, as shown in Author response image 1. The first one is ND98 and we then changed the solution to NMDG-based bath solution. Since the experiment was done in the same cell, to avoid an excessive fluorescence bleaching we repeated the recordings for averaging only 5 times for each of the condition. We applied 300µM ATP to the oocyte in ND98 solution. The non-linear F changes were observed with a significant ATP-evoked current. Then we changed the solution to NMDG-based solution while maintaining the ATP concentration and see if the non-linear F changes would still be observed even though there was less current flow. After changing the bath solution to NMDG, there was a significant reduction of the current, albeit the total reduction of the current could not be achieved. Although the current was significantly reduced, a non-linear F change could still be observed at a similar level. These results suggest that the non-linear F change is not the result of the current flow, but that the current flow is a result of the conformational changes shown by the presence of non-linear F change in NMDG solution.

We also performed the same series of experiments by using mannitol-based bath solution, instead of NMDG, to minimize the ionic current as shown in Author response image 1. The non-linear F changes were observed with a significant ATP-evoked current in ND98 solution. Then we changed the solution to mannitol-based solution while maintaining the ATP concentration. Although there was a significant reduction of the current, a non-linear F change could still be observed. These results further confirmed that the non-linear F change is not the result of the current flow.

**Author response image 1. sa2fig1:** 

After a careful examination of the represented traces shown in figure 4 and figure 6, I am not sure one should directly compare the ΔF/F in the absence and presence of ATP to make the conclusion that "the focused electric field at the position of A337 is stronger in the absence of ATP than in the presence of ATP." (Line 491-492) The noise of the fluorescence signal is always noticeably smaller in the presence of ATP suggesting a possible change in the F. In the hP2X3 receptor closed state structure, residue equivalent to A337 is more than 5Å from that equivalent to F44 in TM1. This part of TM2 is separated from TM1 and Anap incorporated at A337might be surrounded by some lipids in the closed state. In the presence of ATP, when channel opens, this part of TM2 gets close to TM1, where A337 and F44 can interact. Anap incorporated at A337 could be shielded by TM1 in the open state. As one can imagine, those structural rearrangements during channel opening could very likely change the environment for Anap at A337. It would not be meaningful to directly compare ΔF/F recorded in different environment where the fluorophores have inconsistent behavior. I do understand VCF recording was repeated 20 times for each sample in the presence of ATP and only 5 times in the absence of ATP. The difference of noise could also result from the average of different number of traces. Either way, I would suggest the authors to address the difference.

To address whether F intensity significantly changed due to the difference of the environment surrounding A337 in the closed and open state, we measured the F intensity in the absence and in the presence of ATP. This recording was performed in the same cell and each recording for averaging was repeated equally 5 times. The noise level is similar, showing that the apparent difference of the noise level in Figures 4 and 6 is due to the difference of the averaged numbers, as the reviewers pointed out. There was a slight reduction, not increase, in the F intensity in the presence of ATP, which could be due to the fluorescence bleaching, as shown in Figure 6—figure supplement 1. Thus, a larger ∆F/F in the absence of ATP is thought not to be due to smaller F intensity, showing fluorescence change is actually larger in the absence of ATP.

On the other hand, we agree with the reviewers that there is a possibility that the environment surrounding the fluorophore at A337 is different in the closed and open state. A homology structure model of P2X2 also supports that the crevices are different in the closed and open states. It is natural to assume that the environment of the fluorophore is different in the closed and open state (Figure 8C, D). Thus, we agree with the reviewers that we can’t simply conclude that larger fluorescence change means stronger focus electric field. The interpretation of this part is that the focused electric field is present both in the absence and presence of the ATP.

We have now added Figure 6—figure supplement 1 to the revised manuscript and revised the result and Discussion section as follows:

Results section:

“The changes exhibited fast kinetics and a linear voltage-dependence. ∆F/F in the absence of ATP was larger than that in the presence of 10 µM ATP (∆F/F = 0.7%±0.1 at 440 nm (n=4) Figure 4A – C).” We omitted the sentence (line 228-230 in the previous version of the manuscript): “Thus, the focused electric field at the position of A337 is stronger in the absence of ATP than in the presence of 10 μM ATP.”

“Thus, the ∆F in 0 ATP observed in the above experiments might just represent the ∆F in the open state.”

“This mutation was introduced into A337Anap (A337Anap/R313F or A337Anap/R313W) to determine whether the ∆F related to the focused electric field is present at A337, even when the channel is mostly closed in 0 ATP.”

“These results confirmed that ∆F/F at the position of A337 is larger in the absence of ATP than in the presence of ATP. It was of interest to know whether or not the concentration of ATP affects ∆F/F at A337

We revised sentences in Line 260-263 in the previous version of the manuscript:

From (original): “Taken together, these results show that the focused electric field at A337 is [ATP]-dependent and stronger in the absence of ATP, suggesting that the rotation of TM1 upon ATP binding (Figure 8) would tighten the space surrounding A337 making the electric field more converged.”

To: (Line 278-283)

“Taken together, these results show that the ∆F/F at A337 is [ATP]-dependent and larger in the absence of ATP. The higher ∆F/F implies stronger focused electric field in 0 ATP, but it is also possible that the difference comes from the difference of the environment of the fluorophore, as discussed later (Figure 6—figure supplement 1). In any case, it was shown that a ∆F/F and thus a focused electric field are present both in the absence and the presence of ATP.”

Line 374-388:

“Additionally, results consistent with Figure 4E, F, H, and I were also obtained in terms of the fluorescence intensity change of the fast component. ∆F_Fast_/F in the absence of ATP was larger than in the presence of ATP (∆F_Fast_/F=4.4%±0.5 at 440 nm, n=6 and ∆F_Fast_/F=1.7%±0.3 at 440 nm, n=6; Figure 6B). However, there was a concern that F itself significantly changed due to a difference in the environment surrounding A337Anap in the closed and open states. We performed experiments to address this by measuring the absolute F (output of the photomultiplier tube) in the absence and in the presence of ATP. Using the K308R/A337Anap construct, each recording was repeated 5 times, for averaging, in the same cell as shown in Figure 6—figure supplement 1. There was a slight reduction, not increase, in the F in the presence of ATP, which could be due to fluorescence bleaching (Figure 6—figure supplement 1A, B). Thus, a larger ∆F/F in the absence of ATP is thought not to be due to less F, and we concluded that ∆F/F was larger in the absence of ATP. The noise level is similar (Figure 6—figure supplement 2C), showing that the apparent larger noise level in 0 ATP (Figure 4C, 4F, 4I; Figure 4—figure supplement 1H, K; Figure 6B) is not due to smaller F but due to a lower number of averaged traces.”

And the discussion was revised as follows:

Line 460 in the previous version of the manuscript, this sentence was omitted: “The focused electric field was more prominent in the absence of ATP”

Line 512-518 (in the previous manuscript started at Line 458-460):

“We observed that ∆F_Fast_/F changed with voltage in both the closed and ATP bound-open states, implying the presence of the focused electric field in both states at the position of A337. ∆F_Fast_/F was larger in the absence than in the presence of ATP. One simple interpretation is that the focused electric field is stronger in the absence of ATP. However, as the environment of the fluorophore is different in the absence and presence of ATP, due to the difference in the crevice shape (Figure 8C, D), it is not possible to directly compare the strength of the focused electric field in the two states.”

Revision for main figure 8:

We added Figure 8C, D in the revised manuscript (Figure 8), with the intention to make the reader understand that the crevices are different in the closed and open state which resulted in the difference of the focused electric field in both states.

7. The small size of the slow ΔF shown in figure 5 could potentially lead to inaccurate data analysis. It would be better if there are more evidence to support that the slow ΔF results from voltage dependent conformational change in the protein. As previously reported by the Kubo lab, the voltage dependent activation time constant varies with different ATP concentrations. It will be a great supporting evidence if the time constant of the slow fluorescence change vary with ATP concentration.

Thank you for this suggestion. It is also our interest to see the correlation between [ATP] concentration and F change in this study. We have performed VCF recordings of K308R/A337Anap in different [ATP] previously (30 µM; 100 µM; and 300 µM; data for 30 µM and 100 µM ATP was not shown in the manuscript). However, as the reviewer expressed a concern, the data analysis to compare the activation time constant of the slow F change could not be performed reliably, due to a small slow component F change and noise which made it difficult to fit an exponential function.

Nevertheless, the results of the analysis are shown in Figure 5—figure supplement 2E-2G. Current traces could be well-fitted with a single exponential function (Figure 5—figure supplement 2E-F). However, as to the fluorescence traces, as shown in the Figure 5—figure supplement 2E-F, the slow fraction of F change could be fitted at -160 mV but not reliably at -140 mV or -120 mV. The analysis for -160 mV showed that the time constant of the voltage-dependent F change varied with [ATP]. The time constant of voltage dependent F change was larger in the lower [ATP], as depicted in Figure 5—figure supplement 2G. Statistical analysis (unpaired student t-test) showed that the time constant was significantly larger in 30 µM ATP. The larger τ of ∆F/F that than of current at -160 mV in 30 µM ATP (Figure 5 —figure supplement 2E) is puzzling, but we didn’t pursue this point further because of the uncertainty of the fitting.

We have now added Figure 5—figure supplement 2 to the revised manuscript. And revised the result section as follows:

Results:

Line 324-345:

“It was of interest to see the correlation between [ATP] and the ∆F_Slow_/F in this study. We performed additional VCF recordings of K308R/A337Anap in the presence of 30 µM ATP (Figure 5—figure supplement 2A). In the presence of 30 µM ATP, hyperpolarization again elicited fluorescence signals which consist of two components, a very fast decrease (∆F_Fast_/F) and a slow increase (∆F_Slow_/F) to steady-state (∆F_Steady-state_/F) (Figure 5—figure supplement 2A). Plots of the F-V relationship at the end of the recording time interval (at the steady-state) again showed that F-V consists of mixed components, a linear component and a non-linear component (Figure 5—figure supplement 2B). The F-V relationship of ∆F_Fast_/F again showed a linear voltage-dependence (Figure 5—figure supplement 2C), and the F-V relationship of ∆F_Slow_/F showed a non-linear voltage-dependence (Figure 5—figure supplement 2D), similar to the case in 300 µM ATP (Figure 5).

We were curious to know whether the time constant of ∆F_Slow_/F varied with [ATP]. We then analyzed the time constant of activation of the ∆F_Slow_/F in 30 µM ATP and 300 µM ATP. However, the analysis of the activation time constant of ∆F_Slow_ could not be performed reliably, due to a small ∆F_Slow_ component and noise which made it difficult to fit an exponential function. Nevertheless, the results of the analysis are shown in Figure 5—figure supplement 2E-2G. Current traces could be well-fitted with a single exponential function (Figure 5—figure supplement 2E-F). However, as to the fluorescence traces, as shown in Figure 5—figure supplement 2E-F, ∆F_Slow_ could be fitted at -160 mV but not reliably at -140 mV or -120 mV. The analysis for -160 mV showed that the time constant of the voltage-dependent ∆F varied with [ATP]. The time constant of voltage dependent ∆F was significantly larger in the lower [ATP], as depicted in Figure 5—figure supplement 2G.”

8. Why was I341 not studied further? If it is because of technical difficulties with small signals, then we cannot rule out its role in voltage sensing and it should be discussed. Additionally, previous findings from Kubo's lab have demonstrated that mutations at T339 and G342 can completely abolish the voltage dependence, suggesting possible involvement of those residues as well.

[I341] Thank you for addressing this concern. As the reviewer commented, I341 was indeed not studied further due to the technical difficulties with small signals. The ∆F/F was only 0.6%±0.2 at 440 nm with n=3 in Figure 3E even after the application of SIK inhibitor. We also performed the VCF recording in the additional mutation of K308R into I341Anap but the data was not shown in the manuscript due to the small signal which hindered the detailed analysis. Here we show the representative traces of the current and fluorescence from K308R/I341Anap in Author response image 2. F change at the steady-state was smaller compared to K308R/A337Anap (1.7±0.1 at 440nm (n=12) compared to 3.4%±0.3 at 440 nm (n=8)). The F_slow_ component was smaller than that in K308R/A337Anap making it hard to separate the two components. Nevertheless, the addition of K308R mutation also showed there is a minor F slow component detected at I341Anap. Since the component was very minor, we concluded that the main reporter for the voltage-dependent rearrangements of P2X2 receptor is at A337 and the main focus for the voltage-sensing mechanism could lie at the surrounding environment of A337.

Secondly, we have also introduced several single amino acid mutations at I341 position. The results of mutations were not as dramatic as those at F44 or A337; as shown in Author response image 3.

**Author response image 3. sa2fig3:** 

Nevertheless, we certainly agree with the reviewers’ comment that we cannot rule out the role of I341 in voltage-sensing since it also showed detectable but smaller F change. I341 was also shown to be modified by both MTSET and Ag^+^ (Li et al., 2008; Li et al., 2010), suggesting that a narrow water-phase penetrates down to I341. I341 might contribute to voltage-sensing mechanism as one of the focused electric fields where the voltage sensing by F44 takes place.We have now revised the manuscript as follows:

“A337 Cys mutants were first reported to be not modified by MTSET both in the presence or absence of ATP, indicating that this residue is not involved either in the pore lining region in the open state or in the gate of P2X2, unlike I341 Cys mutants, which were modified by MTSET (Li et al., 2008). Meanwhile in another study using Ag^+^, a smaller thiol-reactive ion with higher accessibility, A337C was modified both in the absence and presence of ATP, as well as I341C (Li et al., 2010). These results suggest that a narrow water-phase penetrates down to these positions, which is consistent with the results in this study that there is a focused electric field at the position of A337 and I341.”

[T339] In addition to that, for the previous findings from Kubo-Lab regarding the possible involvement of T339. We observed a very small upward fluorescence change (∆F/F=0.4%) in response to hyperpolarization at the time of the mutant screening but the change was only observed at -140mV, in 1 oocyte out of 5 oocytes tested (The incidence of fluorescence change detection in one batch = 20%). The two traces from different batch are shown in Author response image 4.

**Author response image 4. sa2fig4:** 

We performed further extensive VCF recordings of T339Anap in total 12 batches of oocytes, but was not able to reproduce the result (The incidence of fluorescence change detection between batches was 16.7%). Therefore, it was stated that there was no observed ∆F in response to hyperpolarization in Supplementary Table 1 (Supplementary file 1 in the revised manuscript).Even though we were not able to reproduce the result, we cannot rule out the possible role of T339 in voltage sensing as the mutation to Anap could abolish the voltage-dependent current as well. On the other hand, if we look from the structural point of view, the side chain of T339 is facing to the pore of P2X2 since this is one of the pore lining residues. Therefore, even if T339 contributes to voltage sensing mechanism, we think that the contribution would be indirect.

We also introduced the mutation T339S which had been reported to make the channel constitutively active into K308R/A337Anap to see whether or not the slow component would be diminished. The phenotype of T339S/K308R/A337Anap didn’t resulted in a constitutively active mutant, as shown in Author response image 5, consistent with what had been reported before that K308R mutation diminished the spontaneous activity of T339S (Cao et al., 2007). By comparing with K308R/A337Anap mutant in the same batch, we found that the fluorescence change was significantly reduced by the addition of T339S in the similarly high expression density cells in the same batch. We also found that T339S might reduce the slow fraction of F change at K308R/A337Anap.

**Author response image 5. sa2fig5:** 

[G342 and G344] The expression level of G342Anap was very low and F changes could not be detected as shown in the manuscript Figure 1—figure supplement 3O. It has been reported that the activation phase evoked by hyperpolarization would be mediated by the flexible G344 located in the middle of TM2 (Fujiwara et al., 2009). G344Anap still showed voltage dependence phenotype (Figure 1—figure supplement 3Q in the manuscript) but F change could not be observed even after the application of SIK inhibitor (data not shown).We also introduced the mutation of G344A on top of K308R/A337Anap. G344 is an important hinge which mediates the voltage-dependent activation phase, as reported in the previous study (Fujiwara et al., 2009) that the G344A mutation makes the channel constitutively active. We expected that G344A/K308R/A337Anap will show constitutive activity but the triple mutant responded normally upon ATP application, as shown in Author response image 6. Interestingly, in this construct the fluorescence change was observed even in the case of lower current density. F slow component was still remained.

**Author response image 6. sa2fig6:** 

9. Please provide numbers for the readers to understand how many oocytes were tested and how many showed the change for the statement: "the incidence of fluorescence change detection is also low" (Line 120). The numbers should also be provided later when the authors talk about increase in the incidence of detectable ΔF/F after SIK inhibitor treatment.

Thank you for this suggestion. We have provided the numbers of how many oocytes were tested in the revised manuscript as follows:

“(1) ∆F is close to the limit of detection because signal to noise ratio is low, making it hard to perform further analysis e.g., ∆F/F-V relationships; (2) The incidence of fluorescence change detection in each batch is also low, 14.3±4.1% (n=5-16) and 16.02±0.6% (n=6-13) for A337Anap and I341Anap, respectively. Three out of 13 batches showed F change for A337Anap and 2 out of 10 batches showed F change for I341Anap.”

The detailed numbers of oocytes and batches tested are available in Figure 1—source data 1 (without SIK inhibitor treatment) and Figure 3—source data 1 (with SIK inhibitor treatment).

We have added this incidence of F change detection to the Results section when comparing directly the condition without SIK inhibitor treatment and with SIK inhibitor treatment in the same Anap mutant.

The revisions are as follows:

In A337Anap (Line 173-175):

“Application of 300 nM SIK inhibitor increased the incidence of ∆F (68.8±3.2% (n=6-9) with inhibitor, vs. 14.3±4.1% (n=5-16) without inhibitor) in A337Anap, with an improved signal-to-noise ratio (∆F/F = 1.5%±0.2 at 440 nm (n=8), Figure 3A).”

In I341Anap (Line 210-213):

“Similarly, the application of 300 nM SIK inhibitor to I341Anap resulted in a clearer (∆F/F = 0.6%±0.2 at 440 nm (n=3), Figure 3E) and more frequent Anap ∆F/F (38.1±9.2%; n=5-9), compared to that without SIK inhibitor application (16.02±0.6% (n=6-13)) upon voltage step application in 10 µM ATP.”

10. Is there any correlation between the current size and the detectable ΔF/F? Is it because the channel density is too high for some oocytes to show any voltage dependent change?

There is no clear-cut answer for addressing this matter because detected ΔF/F depends on many aspects. In principle, the smaller the current would be resulted in better ΔF/F because low density means better voltage-dependence at hyperpolarized potentials. However, ΔF/F also strongly depends on the endogenous background fluorescence. If the background fluorescence is relatively high, the ΔF/F could be masked. This is the reason why high expression is needed for successful detection. How high expression level is needed depends on how high the background fluorescence is. It varies from batch to batch and we cannot explicitly state the correlation of the current amplitude and ΔF/F.

11. There is no information about the dose-response of ATP activation for any of the construct tested. What is the rationale for choosing the concentrations used in the experiments?

For the Anap mutant screening experiments, low concentration of ATP (10 µM) was used since G-V will shift to the right as the [ATP] goes higher (Fujiwara et al., 2009). In some cases, the IC50 of P2X2 WT (30 µM) was also used. In special cases the saturating concentration of ATP for WT (100 µM or 300 µM) was used depending on the mutant’s phenotype.

In the case of K308R/A337Anap mutant, it was reported that K308R has a lower ATP sensitivity (Jiang et al., 2000; Cao et al., 2007; Keceli and Kubo, 2009). Therefore 300 µM ATP was used for this mutant.

12. It's shown on Supplementary Table I that no voltage-dependent F change was detected for any residue in TM1 but "n.d." was used to indicate the ATP evoked F change for six residues. What does "n.d." mean exactly and why couldn't it be determined?

Thank you for pointing this out. It is our mistake to mix up the meaning of n.d. for the indication of the ATP-evoked F change for six residues in TM1 as well as 10 residues in the extracellular linker with other n.d. indications on the table. The latter, which are mainly used in intracellular regions, stated that subsequent step of VCF recording could not be performed due to fast current decay.

On the other hand, n.d. indication in ATP-evoked F change in six residues at TM1 (and 10 residues in extracellular linker) simply means not determined / not performed. It is because the main focus of this study was not to observe ATP-evoked fluorescence change but to observe voltage-dependent changes. At the screening process the experiments to determine if there is ATP-evoked change was omitted.

To avoid any confusion, we revised the indication for the Supplementary Table 1 (Supplementary file 1 in the revised manuscript) as follows:

“Mutations were introduced one at a time into 96 positions within the extracellular domain (ECD) near the ATP-binding site and extracellular linker, transmembrane domains (TMs), intracellular N-terminal, and intracellular C-terminal. ATP application ranging from 10 µM, 30 µM, or 100 µM unless otherwise stated. (+) indicates there was either ATP-evoked fluorescence (F) signal change, voltage-evoked F change, ATP-evoked current (I) change, or voltage-evoked I change. (-) indicates negative results. (**) indicates mutants which have a very low expression level, so that the reliable VCF analysis could not be undertaken. (***) indicates fast current decay. (--) indicates that the subsequent recording could not be performed, as a result of fast current decay. (n.d.) indicates not determined.”